# User Experience Evaluation in Intelligent Environments: A Comprehensive Framework

**Stavroula Ntoa** [1,*] **, George Margetis** [1] **, Margherita Antona** [1] **and Constantine Stephanidis** [1,2]

1   Foundation for Research and Technology Hellas, Institute of Computer Science, N. Plastira 100,
    Vassilika Vouton, GR-700 13 Heraklion, Greece; gmarget@ics.forth.gr (G.M.);
    antona@ics.forth.gr (M.A.); cs@ics.forth.gr (C.S.)
2   Department of Computer Science, University of Crete, GR-700 13 Heraklion, Greece
*   Correspondence: stant@ics.forth.gr

**Abstract:** 'User Experience' (UX) is a term that has been established in HCI research and practice, subsuming the term 'usability'. UX denotes that interaction with a contemporary technological system goes far beyond usability, extending to one's emotions before, during, and after using the system and cannot be defined only by studying the fundamental usability attributes of effectiveness, efficiency and user satisfaction. Measuring UX becomes a substantially more complicated endeavor when the interaction target is not just a technological system or application, but an entire intelligent environment and the systems contained therein. Motivated by the imminent need to assess, measure and quantify user experience in intelligent environments, this paper presents a methodological and conceptual framework that provides concrete guidance for UX research, design and evaluation, explaining which UX parameter should be measured, how, and when. An evaluation of the framework indicated that it can be valuable for researchers and practitioners, assisting them in planning, carrying out, and analyzing UX studies in a comprehensive and thorough manner, thus enhancing their understanding and improving the experiences they design for intelligent environments.

**Keywords:** user experience; evaluation; framework; methodology; intelligent environment; smart environment; ambient intelligence; artificial intelligence

## 1. Introduction

Intelligent environments are an emerging field of research and development, constituting a new technological paradigm. The notion of intelligence (both in Ambient and Artificial Intelligence), is becoming a de facto key dimension of the Information Society, since digital products and services are explicitly designed in view of an overall intelligent computational environment [1]. Current technologies are already equipped with intelligence, aiming to support personalization and adaptation to user needs, acting as a stepping stone toward a near future when technology will be omnipresent, machines will predict and anticipate human needs, and robotic systems will be used for various daily activities [2].

In brief, the objective of intelligent environments is to support and empower users; as such, a main thrust of research should emphasize whether and how this goal is achieved, while in this context it is important to consider the implications of user evaluation [3]. Evaluation is a core concern in HCI, with the concepts of technology acceptance, usability and user experience (UX) evaluation constituting the focus of many research efforts that aim to provide answers to what makes a technology usable and acceptable, and the entire experience of using it positive. Although the notions of technology acceptance and usability are not novel, it is notable that as technology moves beyond the typical desktop paradigm, they still constitute objective of active research, through the development of methods, tools, and theoretical frameworks to assess them.

Technology acceptance is defined by two principal factors, namely perceived ease of use and perceived usefulness [4]. However, as technology has evolved from the typical personal computer (PC) to smartphones, tablets, and microcomputers hidden in various devices, while its usage has expanded from the typical workplace domain to several contexts (e.g., household, health, learning, AAL), several other factors have been identified which impact the aforementioned two main factors and eventually technology acceptance. A recent review of 43 relevant models [5], identified 73 parameters influencing technology acceptance, the majority of which (98.92%) is assessed in the various studies through questionnaires, asking users to self-report their characteristics, attitudes and perceptions.

Usability is also fundamental in HCI and an essential component of UX [6]. Since the very first definitions of usability until now, several methods have been proposed aiming to assess the usability of a specific product or service, however, studies have identified that two methods are most commonly employed in usability evaluations, namely user testing and expert-based reviews [7]. With the aim to identify how usability should be measured, several frameworks have been proposed in literature, often influenced by the UX notion [8], and incorporating attributes such as quality in use, societal impact, aesthetics, usefulness, and usage continuance intentions, resulting in a breadth of parameters that should be studied. User Experience (UX) has recently predominated the usability concept, providing a broader perspective on a user's experience with a product, aiming, according to the related ISO standard, to study "a person's perceptions and responses resulting from the use and/or anticipated use of a product, system or service", and including all the users' emotions, beliefs, preferences, perceptions, physical and psychological responses, behaviors and accomplishments that occur before, during and after use of a product or service [9]. UX methods that go beyond usability evaluation are mainly focused on users' perceptions of system attributes (e.g., aesthetics, playfulness, and fun), as well as on the emotions induced by system usage. In an effort to provide a more systematic approach towards assessing UX, several frameworks have been proposed, the majority of which have, however, remained conceptual.

At the same time, intelligent environments impose novel challenges to the evaluation of UX. Such challenges pertain to the nature of interaction, which shifts from explicit to implicit, encompasses novel interaction methods, and is escalated from one-to-one to many-to-many interactions [2]. At the same time, intelligent environments besides human–thing interactions also encompass 'thing-to-thing' interactions, which introduce additional concerns regarding conflicts' resolution, interoperability, and consistency of interactions [10]. To this end, several efforts have attempted to "frame" evaluation and define how it should be pursued in terms of usability, user experience, as well as interaction adaptation and ubiquitousness. Nevertheless, as technology advances, the number of parameters to be assessed becomes too large to be studied through user experiment observators' notes, or evaluation questionnaires to be filled-in by users (a common current practice when evaluating user experience). On the other hand, despite the fact that the notion of intelligent environments has existed for more than a decade and the vital importance of evaluation, efforts in the domain have mainly focused in identifying the challenges in the field and advocating the importance of in situ evaluations, while there is a lack of generic and systematic approaches towards user experience evaluation in such environments.

Motivated by the need to define how user experience should be assessed in intelligent environments, as well as by the general lack of approaches with practical value in the field of evaluation frameworks, this paper proposes a novel comprehensive framework, named UXIE, for the evaluation of User Experience in intelligent environments (IEs), aiming to assess a wide range of characteristics and qualities of such environments, taking into account traditional and modern models and evaluation approaches. The proposed framework adopts an iterative design approach, suggesting specific evaluation approaches for the different development stages of an intelligent environment, system, or application, thus allowing the assessment of the user experience from the early stages of the development lifecycle to the final stages of implementation. UXIE is a clean-cut conceptual and

methodological framework, taking into account the various facets and temporal attributes of UX, providing not only concepts, but also concrete metrics and methods to measure them. Furthermore, taking advantage of intelligent environments' architecture and sensors' infrastructure, it advocates the automatic identification of specific metrics, alleviating the need for observers to keep lengthy notes or to address all issues through questionnaire items to be answered by users. The main contribution of this paper is the framework itself, which goes beyond existing approaches in providing not only concepts that should be evaluated, but also specific metrics, and identifies the methods that should be used. The proposed framework encompasses a total of 103 concrete metrics, 41 of which are novel. At the same time, novel UX dimensions of IEs are explored through the proposed framework, such as adaptation impact, implicit interactions, and actual usage of an application or service. Furthermore, the proposed framework introduces the concept of automated measurements, derived through the IE, thus increasing the objectivity of measurements and minimizing the evaluator's effort.

The remainder of this paper is organized as follows. Section 2 discusses related work and introduces relevant frameworks. The methods and materials that were employed in order to reach the proposed framework, besides literature review, are discussed in Section 3. The proposed framework itself is described in detail in Section 4. Section 5 presents the evaluation of the framework in terms of methods, procedures, and results. Discussion on the findings is conducted in Section 6, whereas Section 7 concludes the paper and highlights directions for future work.

## 2. Related Work

Although UX is a relatively young field and a challenging subject to define and understand, several frameworks have already been proposed for its evaluation. However, most of the proposed frameworks in literature are mainly conceptual or focus on specific contexts and application domains.

A conceptual framework that has been proposed as a medium to design and evaluate UX is that of Hellweger and Wang [11]. According to the framework, UX is affected by six prime elements, namely context, usability, product properties, cognition, needs, and purpose. UX produces the following six prime elements that should be pursued and assessed: memorability, ubiquity, perception, emotional state/mood, engagement, and educational value. For each of the 12 prime elements, the framework describes subelements that should be taken into account, resulting in 86 attributes, which however are not further defined as to how they can be assessed (e.g., efficiency, behavior patterns, perceived quality, etc.).

Miki [12] proposed an integrated evaluation framework of usability and UX, according to which during use and after use measurements are proposed. During use measures are further analyzed into objective measures of usability and more specifically effectiveness and efficiency, as well as subjective measures of UX, namely perceived quality, perceived value and satisfaction. After use measures pertain only to UX and include complaints and customer loyalty.

The QUX tool was proposed to support a common organizational understanding of a product's UX and the selection of further in-depth UX evaluations [13]. The tool encompasses 28 consolidated UX characteristics under seven main clusters, as follows: (1) emotion: satisfaction, pleasure; (2) design: interface, aesthetics; (3) content: information, effectiveness; (4) technology: efficiency, functionality, ease of use, performance, usability, utility, security, control, learnability; (5) result: quality of outcome, error-free; (6) further disciplines: brand history, advertisement, price, user expectation, user customization, user self-realization, group affiliation, social connectivity; (7) environment: memorability, time context, location context. This list of UX characteristics was further analyzed and clustered under the categories of look, feel, and usability, reaching nine UX dimensions, each explored through three questions that a user will have to answer. The UX dimensions studied were appealing visual design (look), communicated information structure (look), visual

branding (look), mastery (feel), outcome satisfaction (feel), emotional attachment (feel), task effectiveness (usability), task efficiency (usability), stability and performance (usability).

Another more recent UX evaluation framework [14] involving different UX aspects and dimensions, and measurement methods, encompasses three core dimensions, namely user needs experience, brand experience, and technology experience which shape the overall value of a product or service. These are further analyzed to categories and contexts, resulting in the definition of the following UX aspects: visual attractiveness, platform technology, infrastructure, service response time, business communications, marketing, everyday operations, functionality, usability, usefulness, sensual, pleasure/fun, emotional, trustworthiness, and aesthetics.

Several efforts have focused on the evaluation of ubiquitous computing environments, which can be seen as predecessors of IEs. The origins of ubiquitous computing can be attributed to Mark Weiser, who described his vision for the 21st century computing stating that: "The most profound technologies are those that disappear. They weave themselves into the fabric of everyday life until they are indistinguishable from it" [15]. Ubiquitous computing is the term given to the third era of modern computing, which is characterized by the explosion of small networked portable computer products in the form of smart phones, personal digital assistants, and embedded computers built into many of the devices we own—resulting in a world in which each person owns and uses many computers [16].

Building on the technology acceptance model [4] and its extensions, and further advancing them to address pervasive and ubiquitous environments, the pervasive technology acceptance model, named PTAM, was introduced [17]. As an alternative to evaluating a pervasive computing application, in situ or in the laboratory, the model aims to predict user acceptance and long-term usage after minimal exposure to a prototype. PTAM added the following constructs to previous approaches: (i) trust, which in pervasive environments is very important due to the nature of data that are collected by the environment; (ii) integration, aiming to assess whether the technology is well-integrated into the environment and does not distract users or interferes with their other activities. Furthermore, it defined usage motivation and socioeconomic status as motivators, along with other user attributes, such as gender, age, and experience. Given the large corpus of research related to technology acceptance models, most of the parameters that PTAM introduced have already been addressed in other models. Nevertheless, the construct of integration is very important for ubiquitous environments. Yet, the framework did not include specific suggestions on how to measure the integration construct and has not been validated.

A framework for ubiquitous computing evaluation, defining a set of evaluation areas, sample metrics and measures was developed by Scholtz and Consolvo [18]. In more detail, the following nine evaluation areas and their related metrics are foreseen by the framework: (i) attention, with metrics focus and overhead; (ii) adoption, which can be measured by rate, value, cost, availability, and flexibility; (iii) trust, with privacy, awareness and control metrics; (iv) conceptual models, measured with the help of predictability of application behavior and awareness of application capabilities; (v) interaction, measured by effectiveness, efficiency, user satisfaction, distraction, interaction transparency, scalability, collaborative interaction; (vi) invisibility, with metrics intelligibility, control, accuracy and customization; (vii) impact and side effects, measured through utility, behavior changes, social acceptance, and environment change; (viii) appeal, with metrics fun, aesthetics, and status; (ix) application robustness, with metrics robustness, performance speed and volatility.

A recent framework was proposed [19], featuring a set of 27 quality characteristics that should be considered for the evaluation of ubiquitous computing systems, namely acceptability, attention, availability, calmness, context-awareness, device capability, ease of use, effectiveness, efficiency, familiarity, interconnectivity, mobility, network capability, predictability, privacy, reliability, reversibility, robustness, safety, scalability, security, simplicity, transparency, trust, usability, user satisfaction, and utility. Additionally, a detailed list of 218 software measures to achieve the aforementioned evaluation of quality characteristics is proposed, with an indication of how well they are defined in the referenced sources. It

is notable that out of the 218 measures, only 36 are well defined, and the remaining 182 are either defined but without measurement function or not defined at all. Despite the existence of these frameworks, however, a recent study on the evaluation of ubiquitous computing and Ambient Intelligence environments [20], exploring a total of 548 relevant evaluations, identified that 38.5% of these studies used a standardized UX evaluation questionnaire, whereas out of the 61.5% remaining the majority of studies employed one additional questionnaire, SUS, which is a popular usability evaluation questionnaire. It is noteworthy that the majority of evaluations mainly resorted to subjective metrics provided by users themselves and did not explore additional UX metrics. This can be attributed to various factors, including limited time, and lack of appropriate resources to design and apply a more comprehensive evaluation.

In the context of evaluating UX in IoT environments, the CHASE checklist has proposed assessing via expert-based reviews attributes of such environments that affect user behavior [21]. In particular, the checklist involves 26 points, guiding evaluators towards assessing UX in two categories, namely human–thing and thing–thing interactions, and three domains across these categories, and in particular general UX aspects (such as signs of user contentment or discomfort, recognition of interactive "things", correct manipulation of "things", positive and negative statements, etc.), context awareness, and programmability.

Studying the concept of evaluation of adaptation, which is an inherent feature of intelligent environments, the following usability factors have been identified for the evaluation of interactive adaptive systems: predictability, privacy, controllability, breadth of experience, unobtrusiveness, timeliness, appropriateness, transparency, comprehensibility, scrutability, effectiveness, efficiency, and precision [22]. The methods typically employed in any of the possible adaptation layers are identified to be cognitive walkthrough, heuristic evaluation, focus groups, user-as-wizard, task-based experiments, and simulated users. A taxonomy for the evaluation of adaptive systems is proposed in [23], based on five dimensions: scope, time, mechanisms, perspective and type. A detailed list of attributes for the quality evaluation of such systems is proposed including quality of service, cost, flexibility, failure avoidance and robustness, degree of autonomy, adaptivity, time to adapt, sensitivity, and stabilization, most of which constitute technical aspects of the system which affect the overall UX, yet they are not concrete metrics that can be employed for UX evaluation.

In the context of IEs, very few efforts have focused on providing a framework for evaluation. An example, albeit quite generic and focusing on the processes rather than on the metrics, is Experience Research theory, that supports user-centered design in Ambient Intelligence environments [24]. Experience Research theory involves studies in (i) context, which focuses on collecting initial user requirements without introducing any new technology applications; (ii) the laboratory, with the aim to evaluate the new propositions in a controlled setting; (iii) the field, which allows long-term testing in real life settings. Therefore, three dimensions can be identified in the process of generating experiences for Ambient Intelligence (AmI) environments: Experience@Context, which involves trend studies, insight generation and validation; Experience@Lab, which may encompass concept definition, experience prototyping and user-centered design and engineering; Experience@Field, involving involves field tests, longitudinal studies and trials [25].

Another framework, aimed at assisting the design and evaluation of IEs [26], proposes using storytelling videos to describe and communicate the user values and design scenarios to stakeholders, and then generating design proposals on five factors (context of interaction, required system data, required sensing input, required user input, and desired system output). Evaluation of the proposed solutions targeted at assessing the perceived feasibility and user acceptability, highlighting three main steps towards implementation of the designed solution. The framework describes an interesting proposal for the entire lifecycle of designing intelligent systems; however, it cannot constitute a methodological guide, assisting UX researchers and practitioners in identifying how to measure UX.

A framework oriented toward recognizing the user social attitude in multimodal interaction in smart environments is proposed by De Carolis, Ferilli and Novielli [27].

According to the proposed framework, signals of social attitude in multimodal interaction can be decomposed into signals in language, speech, and gestures. As a result, the user modeling procedure of the framework integrates: (i) language analysis for linguistic cues extraction, (ii) prosodic analysis, and (iii) gesture recognition into a Dynamic Belief Network. At the beginning of interaction, the model is initialized, while at every dialog step knowledge about the evidence produced by the multimodal analysis is entered and propagated in the network, and the model revises the probabilities of the social attitude node. The new probabilities of the signs of social attitude can be used for planning how the environment will behave.

Although the above frameworks introduce concepts relevant to IEs and provide a classification of measures and metrics, in their majority they do not systematically assist evaluators in deciding which evaluation method to choose, or which exact metrics, according to the specific evaluation context (e.g., the context of use of the system evaluated, the development stage of the system, the users or experts that will be involved in the evaluation). On the other hand, given the high complexity of ubiquitous and pervasive computing environments, frameworks often end up with an unmanageable number of parameters, attributes and constructs that should be evaluated. In this respect, following the review of relevant frameworks, taking into account the attributes suggested as well as their shortcomings, and further elaborating on UX aspects that should be studied in intelligent environments, the UXIE framework was developed. The proposed framework aims to bridge gaps of existing frameworks, providing concrete guidance for the evaluation of UX in intelligent environments, and offering not only conceptual, but also methodological guidance, as well as checklists that can be used by evaluators to identify which what to measure and when.

## 3. Materials and Methods

### 3.1. Attributes of Intelligent Environments—Conceptual Overview

Several definitions have been provided for intelligent environments, each describing with slight variations the features of such environments. Overall, such environments have been identified as interconnected, pervasive, transparent and nonintrusive, able to recognize objects and people, learn from their behavior and adapt to support them [25,28–32]. A word cloud presenting all the attributes and characteristics that have been encountered in the various definitions of intelligent environments, according to their frequency of occurrence is presented in Figure 1.

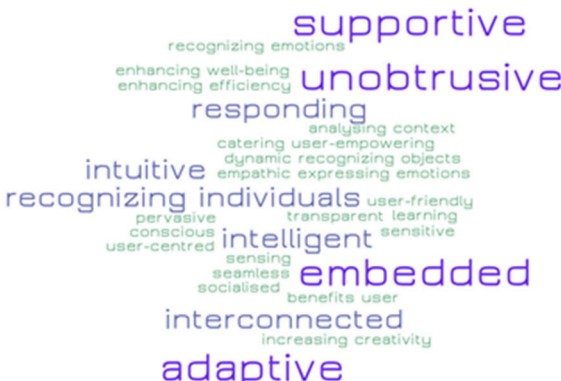

**Figure 1.** Attributes and characteristics of intelligent environments.

Taking into account the various characteristics of intelligent environments, the UXIE framework foresees the evaluation of seven fundamental attributes, namely intuitiveness, unobtrusiveness, adaptability and adaptivity, usability, appeal and emotions, safety and privacy, as well as technology acceptance and adoption (Figure 2). This section describes the UXIE framework from a conceptual point of view, discussing the importance of each of

the seven attributes in the context of the UX evaluation, and presenting the main high-level characteristics that determine each attribute.

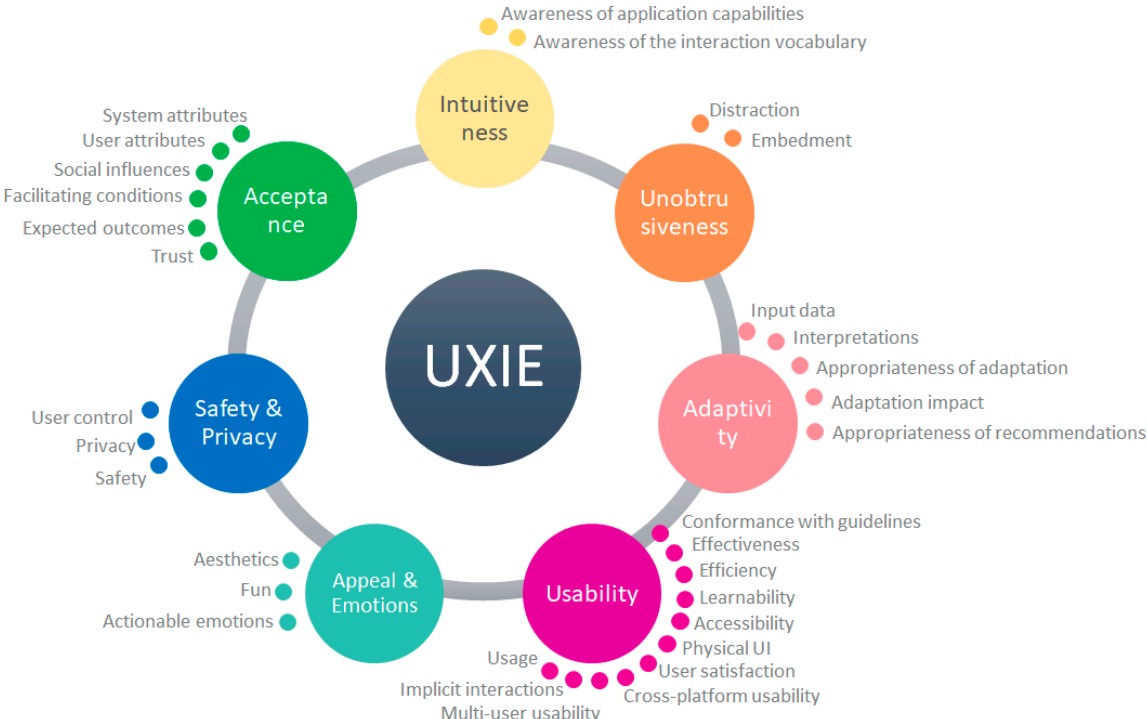

**Figure 2.** Attributes and characteristics of intelligent environments in the UXIE framework.

Intuitiveness and unobtrusiveness are two important characteristics that intelligent environments should exhibit. Intuitiveness is desirable for any system, underpins good design, and in general it means that the system employs pre-existing action–perception (motor) routines and socially (and culturally/historically) acquired "know-how", thus allowing users to focus on achieving a target goal through a system rather than on interacting with it [33]. In the context of IEs, where novel means of interaction are inherently supported, applications may be pervasive, devices interconnected, and the system proactively anticipates and, in some cases, acts on behalf of the user, intuitiveness becomes a major need and challenge. The proposed framework suggests two main characteristics that should be assessed in this direction, namely that users are aware of the application/system capabilities and of the interaction vocabulary. Unobtrusiveness suggests that the system should not obstruct the users' main tasks [34] or generally place demands to the user's attention that reduce the user's ability to concentrate on their primary tasks [35]. As a result, systems comprising the IE should be appropriately embedded in the physical environment, and support user interactions without inducing distractions.

Adaptability and adaptivity are core attributes that deal with the static and dynamic adaptations of the IE according to each different user or user group and context of use. Context of use refers to the devices, the environment characteristics (e.g., light and sound levels) and the domain under which the system is being used (e.g., work, education, leisure, entertainment). Following the layered evaluation approach [34], adaptations are proposed to be studied in different layers, namely regarding the accuracy of data acquired through the environment's sensors, validity of interpretations, and appropriateness of an adaptation studied along three dimensions: interaction modalities supported, output provided and content delivered. The impact of an adaptation should also be explored, referring to how users react once an adaptation has been applied (e.g., if errors are increased). Last, as recommendations are also based on the same layers as adaptations, requiring valid input data, and appropriate inferences based on user and context models, the appropriateness of

recommendations is another system characteristic assessed in the context of adaptability and adaptivity.

The cornerstone of the overall user experience is usability, referring to usability issues of each system in the IE and to the usability of the entire IE, studying cross-platform usability, multiuser usability and implicit interactions, issues that are imperative to be evaluated given the confluence of platforms and systems and the pervasiveness of applications, as well as the multiple users who may interact with the environment explicitly or implicitly, posing sometimes conflicting demands and requirements. Individual systems' usability refers to the qualities of each system that comprises the IE, qualities which allow users to interact with it in an effective, efficient and satisfactory manner, also including learnability, accessibility, and conformance to relevant guidelines. Furthermore, the physical UI design of the individual systems should be assessed, as interaction in IEs goes beyond the typical desktop paradigm to using and interacting with novel objects [36]. As a final point, the actual usage of the individual systems and applications of the IE should be considered, with the aim to identify any usage patterns or preferences, and also detect systems and applications that are not used often or that are used for short periods of time.

Taking into account that user experience goes beyond usability assessment into looking users' emotions, perceptions, as well as physical and psychological responses, the framework includes the attribute of appeal and emotions. To this end, it deals with the aesthetics of the IE and the systems that compose it, assesses how fun the users perceive the IE and/or its systems to be, and how they actually feel. The latter is explored through users' reporting their affective reactions, as well as through detecting potential emotional strain through physiological measurements.

Safety and privacy are important parameters of the overall user experience and user acceptance of any technology. Especially for IEs and given their inherent capability to collect data on people's everyday interactions and to search large databases of collected data, the issue of privacy becomes critical. Under this perspective, the framework studies the control that a user will have over the data that are collected by the environment and the information dissemination (i.e., if and what data will be communicated to other systems), as well as identity security issues. In addition, the level of control that the IE has over the individual should be assessed. Finally, issues related to safety should be taken into account, including commercial damage, operator health and safety, public health and safety, as well as environmental harm.

Last, taking into account the holistic approach of user experience, studying the user's perceptions before, during and after the use of a specific product, the framework caters for studying the overall technology acceptance and adoption of an IE. This can be further analyzed by studying system features as they are perceived by the user, user attributes, and social influences to use a specific system, facilitating conditions, expected outcomes, and trust.

### 3.2. Evaluation Approach—Methodological Overview

A fundamental constraint in existing approaches is that several of the user experience qualities that the evaluation aims to assess are measured through questionnaires, by recording the user's subjective opinion on a matter. As a result, if one would like to study a plethora of issues, the evaluation questionnaire would end up being too large to be answered. To this end, the proposed framework aims to assess as many issues as possible through other methods. However, user testing is the most fundamental evaluation method [37] and cannot be completely replaced by any other method, therefore it constitutes a core evaluation approach of the framework.

Following an iterative approach, the framework proposes a combination of formative and summative evaluation methods, namely expert-based reviews and user testing (Figure 3). These two methods are the most popular and actually employed during evaluations [38]. During the design and prototyping phases of an IE application or system, the framework proposes evaluation through expert-based reviews. As the center of the

iterative design approach is the recurrence of evaluation and the improvement of designs and prototypes based on the evaluation results, expert-based evaluations can be planned by the evaluator when appropriate. Once a fully functional prototype is available, or when the evaluator deems suitable, user-based UX evaluation can take place. It should be noted that the framework describes what should be measured and how, and simply provides suggestions as to when. Evaluators can employ the proposed methods according to their own experience and needs during the lifecycle of the development of an application or system that will be deployed in an IE. Moreover, the scope of the evaluation may vary from a specific application running in one system, to a system including many applications, a pervasive application running in multiple systems, or an entire intelligent environment.

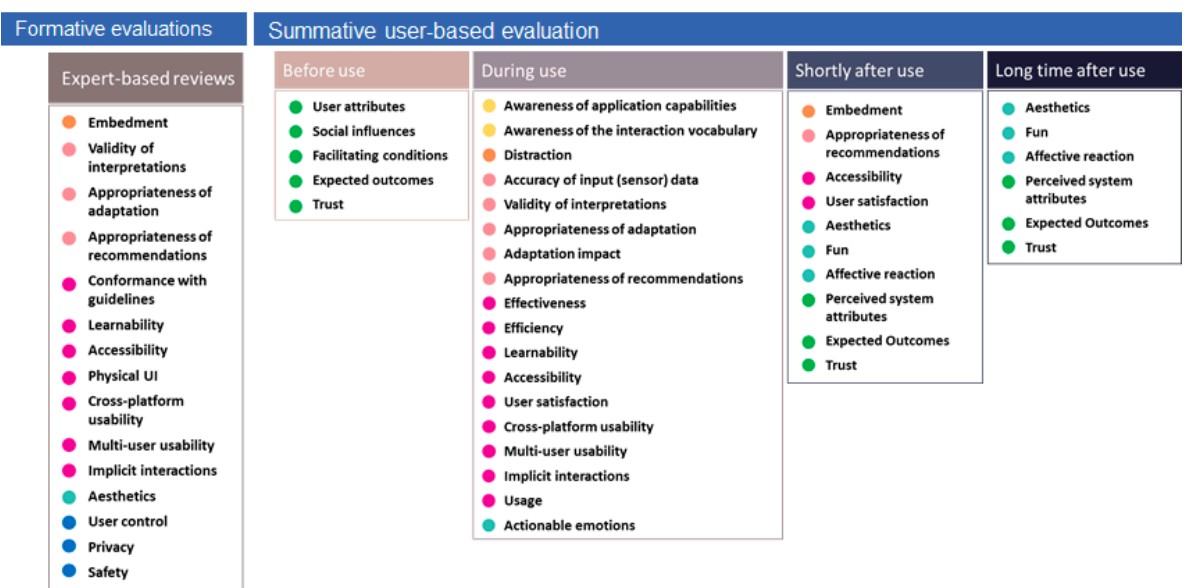

**Figure 3.** Evaluation approaches employed in the context of the UXIE framework.

In a nutshell, UXIE proposes combining formative and summative methods for better results. This combination is common practice in evaluations [38], since through formative evaluations several major problems can be eliminated without the need for involving actual users, or running resource demanding long-term experiments. On the other hand, it has been shown that the different assessment approaches and more specifically expert-based reviews and user-based evaluations find fairly distinct sets of usability problems, therefore they complement each other [37].

In particular, expert-based reviews may be used to assess various aspects of the individual systems in the IE, such as embedment, validity of interpretations, appropriateness of recommendations, compliance with general and domain-specific guidelines, accessibility, physical UI, aesthetics, user control over the data collected and the behavior of the IE, as well as privacy and safety. User testing constitutes a vital approach for the evaluation of user experience in intelligent environments. It should be noted that all the user-testing protocols (e.g., thinking-aloud, retrospective testing, coaching, co-discovery learning, cooperative evaluation, etc.) can be applied, while user testing is used as a term for any type of test that employs users and namely (task based) tests in simulation spaces (Living Labs), in situ evaluations, or real long-term usage in IEs. An important contribution of the framework is that it enhances the evaluation process with automated measurements provided through the environment itself and its infrastructure. Moreover, for the majority of metrics pursued to be recorded through observation, the ones for which automation support through tools is feasible are clearly marked. A combination of the automated measurements, metrics with automation support, user observation, questionnaires and interviews is expected to allow evaluators to gain insight into the composite issue of user

experience. In order to effectively combine all the aforementioned information deriving from different sources, an important concern that should be addressed is that of synchronizing automated measurements, evaluator observations, and video recordings, in order to further assist the evaluator in comprehending interaction difficulties and deriving useful conclusions. This can be achieved through appropriate tools, such as for example UXAmI Observer, which is a tool to support evaluators in carrying out user-based evaluations in IEs or IE simulated spaces. UXAmI Observer aggregates experimental data and provides an analysis of the results of experiments, incorporates information regarding the context of use and fosters the objectivity of recordings by acquiring data directly from the infrastructure of the IE [39].

### 4. The UXIE Evaluation Framework

Having studied the attributes of IE that the framework aims to assess, as well as the evaluation methods that may be employed to this end, this section presents the proposed framework, including metrics and measurement approaches for each attribute.

In the context of intuitiveness, the awareness of application capabilities can be measured by identifying the functionalities that have been used for each system, as well as the undiscovered functionalities. These metrics can be provided automatically by the IE itself with the use of appropriate instrumentation. More specifically, two preconditions need to be met: (i) declaration of the entire set of functionality supported by an application, and (ii) communication of the application with the IE to identify when a specific functionality is used. Awareness of the interaction vocabulary is based on exploring input commands provided by the users, and more specifically: (i) calculating percentages of input modalities used, that is which exact modalities are used by the user in their interaction with the system and how often, highlighting thus users' preferences regarding the supported input modalities, (ii) identifying erroneous user inputs per input modality (e.g., gesture, speech, etc.), and in particular user input commands that have not been recognized by the system, and (iii) percentage of erroneous user inputs per input modality, providing a general pointer as to how easy it is for a user to employ the specific modality. The aforementioned measurements can also be automatically acquired.

With regard to unobtrusiveness, distraction is measured through the number of times that the user has deviated from the primary task, as well as the time elapsed from a task deviation until a user returns to the primary task. Both metrics mainly apply to task-based evaluations or free exploration through thinking aloud, as in free exploration and usage it is not possible to know or to always correctly infer the user's goal, unless explicitly stated by users themselves. Evaluators can be assisted by appropriate tools in calculating these metrics, by having only to mark (e.g., through pressing a specific key) when a task deviation starts and when it ends. The characteristic of embedment, and more specifically whether the system and its components are appropriately embedded in the surrounding architecture, are suggested to be evaluated by experts, as well through questionnaire and interviews with the users after their interaction with the system.

Adaptability and adaptivity are proposed to be evaluated through assessing five main characteristics, following the paradigm of layered evaluation. First, the accuracy of input data perceived by the system should be assessed (e.g., accuracy of the data received by the sensors). This can be carried out through user testing. Automation support can be provided, by displaying to the evaluator all the input data acquired through the environment sensors not in a raw format but elaborated in a semantically meaningful form. The next assessment level refers to the validity of interpretations, a metric which can be calculated through expert-based review of the adaptation logic, and user testing with automation support. Automation support in this case refers to displaying, in a meaningful manner, the specific inferences of the reasoning mechanism, prior to applying an adaptation. At the next level, the appropriateness of an adaptation is evaluated, by means of exploring whether the interaction modalities, the system output, and the content are appropriately adapted according to the user profile and context of use, through user testing with automation

support. The metric of adaptations that have been manually overridden by the user indicates whether an adaptation is not only appropriate but acceptable as well and can be acquired through automation supported user testing. Automation refers to the potential of the environment to detect when a user interaction possibly denotes an objection to the adaptation applied, by changing the state of a system that was also modified in the context of an adaptation (e.g., if the environment dims the lights following a suggestion by a reasoning agent, while the user turns them to full bright). The confirmation of whether the adaptation was actually rejected by the user should be provided by the evaluator. Besides being appropriate and acceptable, an adaptation may impose difficulties to a user, therefore its impact should also be assessed. To this end, the automated measurements of the number of erroneous user input commands once an adaptation has been applied and percentage of manually overridden adaptations can be employed. Additionally, the number of erroneous user interactions (e.g., selecting a wrong menu item) can provide an indication on the impact of the adaptation, which can be automatically calculated based on instances of interaction errors marked by the evaluator. Finally, the appropriateness of recommendations can be assessed through the following metrics: if adequate explanations of any recommendations are given by the system (assessed through user testing with automation support), if it is possible for a user to express and revise their preferences (by expert-based review), if recommendations are appropriate for the specific user and context of use (via expert-based review and user testing with automation support), which specific recommendations have not been accepted by the user (user testing with automation support), percentage of accepted system recommendations (automated measurement in user testing), and finally user's satisfaction by the system recommendations assessed through questionnaire and followed up by interviews if needed.

The next attribute, usability of the specific systems and the entire IE, is studied through the evaluation of 11 characteristics analyzed in specific metrics. The system's conformance with guidelines should be at first evaluated by expert-based review, taking into account all the guidelines that are relevant for the systems and applications under inspection. Effectiveness can be measured by two fully automated metrics, number of input errors and number of system failures, and two metrics with automation support, namely task success and number of interaction errors, where the environment can produce calculations based on actual values indicated by the evaluator. Efficiency is proposed to be measured by the automated metric of time on task, and two metrics with automation support, number of help requests and time spent on errors. Learnability can be evaluated via cognitive walkthrough carried out by experts, as well as by studying users' performance (number of interaction errors and number of input errors) and help requests over time, metrics which can be calculated automatically. Accessibility can be inspected by experts assisted by semiautomated evaluation tools to assess conformance with accessibility guidelines. Accessibility refers both to electronic and physical accessibility and can be assessed both by experts and by user testing, focusing on observations regarding accessibility problems and retrieving users' opinion through interviews. Electronic accessibility deals with the qualities of the software systems that constitute the IE, which should allow their effective and efficient usage by users with functional limitations due to disability or aging [40]. Physical accessibility, on the other hand, refers to the attributes of the environment that constitute it usable by diverse target user groups (e.g., elderly, disabled, children). The overall physical design should be assessed by experts studying whether the system violates any ergonomic guidelines and checking whether the size and position of the system and its interactive controls is appropriate for manipulation by target user groups. The latter can also be explored through user testing by observing users' interaction with the physical elements of systems in the IE. User satisfaction is typically assessed through questionnaires aiming to elicit users' opinions regarding the system. Besides, during a user testing session, the following can be recorded as indicators of user satisfaction: favorable and unfavorable user comments, statements expressing frustration, and declarations of clear joy. Although these need to be manually indicated by the observer, automatic calculation of percentages

and total numbers of the above indicators constitute metrics of user satisfaction. The characteristic of cross-platform usability involves metrics studying consistency among the user interfaces of the individual systems, appropriateness of content synchronization and actions, which can be inspected by experts. Additional metrics refer to user interaction and behavior once the user switches devices (platforms), and in more detail: the time spent to continue the task from where it was left, help requests after switching devices and comparisons of cross-platform task success and task times, for task-based evaluations. All these metrics can be acquired and calculated through user tests, either with automation support or fully automated. In all cases, the environment can effectively detect when the user has changed device, requesting evaluator input only for metrics that cannot be fully automated (e.g., task success). Multiuser usability involves measuring, through automated measurements, the number of collisions with activities of others and conflicts resolved by the system. The evaluator can also observe via user testing and indicate conflicts resolved by users themselves and the correctness of the system's conflict resolution, supported by appropriate tools in calculating total numbers and percentages. Last, experts should carry out inspections of the behavior of the IE to verify that it does not violate social etiquette. Implicit interactions refer to actions performed by the user that is not primarily aimed to interact with a computerized system, but which such a system understands as input [41], and can be explored by reviewing which implicit interactions occur and of what type (e.g., location-based, emotion-based, etc.). It is also important to study the appropriateness of system responses to implicit interactions, a task which can be supported by the environment by displaying all system responses after an implicit interaction, allowing evaluators to assess its appropriateness, and by calculating numeric metrics based on evaluators' judgement. Finally, the metrics regarding the actual system and application usage in the IE, which are all acquired through user testing and are automatically provided by the environment are number of usages per hour on a daily, weekly and monthly basis for the entire environment, as well as for each system and each application; time duration of users' interaction with the entire environment and also with each individual system and application, analysis (percentage) of applications used per system, as well as analysis (percentage) of systems to which a pervasive application is actually deployed.

Evaluation of appeal and emotions involves examining metrics related to aesthetics, fun, and users' emotions. More precisely, aesthetics are evaluated by experts reviewing if the systems follow principles of aesthetic design and reporting any violations, as well as by asking users their opinion on the matter through questionnaires. Fun and users' affective reaction to the systems are also suggested to be assessed by users' responses to questionnaires. Finally, taking into account that physiological measurements can be acquired through sensors of the IE, actionable emotions can be automatically detected and brought to evaluators' attention.

Characteristics and metrics related to safety and privacy are proposed to be evaluated through expert based reviews. In particular, user control can be assessed by verifying that the user has control over the data collected and the dissemination of information, and also that they can customize the level of control that the IE has on behalf of the user (e.g., acts on behalf of the person, gives advice, or simply executes user commands). Privacy involves inspecting the availability of the user's information to other users of the system or third parties, the availability of explanations to a user about the potential use of recorded data, as well as the comprehensibility of the security and privacy policy. Lastly, safety involves inspecting if the IE is safe for its operators and safe in terms of public health, and it does not cause environmental harm or harm to commercial property, operations or reputation in the intended contexts of use.

Technology acceptance characteristics are pursued through users' responses to questionnaires delivered, before, shortly, and/or long after the user's interaction with the system. System attributes aimed to be assessed are perceived usefulness and ease of use, trialability, relative advantage, as well as installation and maintenance cost. Questions regarding cost should not be necessarily addressed to the end-users, as they are not al-

ways the ones directly responsible for it (e.g., in an organizational or public setting). User attributes that should be explored include the user's self-efficacy, computer attitude and personal innovativeness, as well as their age and gender. Metrics regarding social influences include subjective norm and voluntariness, while the ones related to facilitating conditions are end-user support and visibility. Expected outcomes can be explored in terms of perceived benefit, long-term consequences of use, observability, and image. Finally, user's trust towards the system should also be assessed, as it is an important parameter affecting adoption intentions.

All the specific metrics that the UXIE framework proposes, categorized under characteristics and general attributes to be assessed are listed in Table 1, reporting the appropriate methods for each metric. Metrics that are novel in the UXIE framework are identified by an asterisk. Metrics acquired through user testing include the following additional indications:

- Whether automation is possible, with the indication automated measurement for full automation and automation support whenever full automation is not possible, but the evaluator can be assisted in calculations and observation recording. In general, fully automated measurements are based on the architecture of IE and the typical information flow in such environments, whereby interactors (e.g., people) perform their tasks, some of these tasks trigger sensors, and these in turn activate the reasoning system [29]. Therefore, interaction with a system in the IE is not a "black box", instead it goes through sensors and agents residing in the environment, resulting in knowledge of interactions by the environment. A more detailed analysis of how the architecture of IE can be used for the implementation of such automated measurements is provided in [39].
- If the metric should be acquired before the actual system usage (⏱ B), during (⏱ D), shortly after (⏱ sA), or long after it (⏱ lA).
- If the metric pertains to a task-based experiment (Task-based), or if it should be applied only in the context of real systems' usage (e.g., in in-situ or field studies).
- If the metric is to be acquired through a specific question in the questionnaire that will be filled-in by the user after their interaction with the system, or as a discussion point in the interview that will follow up.

Overall, the framework includes 103 specific metrics that can be collected through a combination of methods, as shown in Figure 4. More specifically, 20 metrics are assessed through expert-based reviews, 72 metrics through user testing, and 11 by both methods.

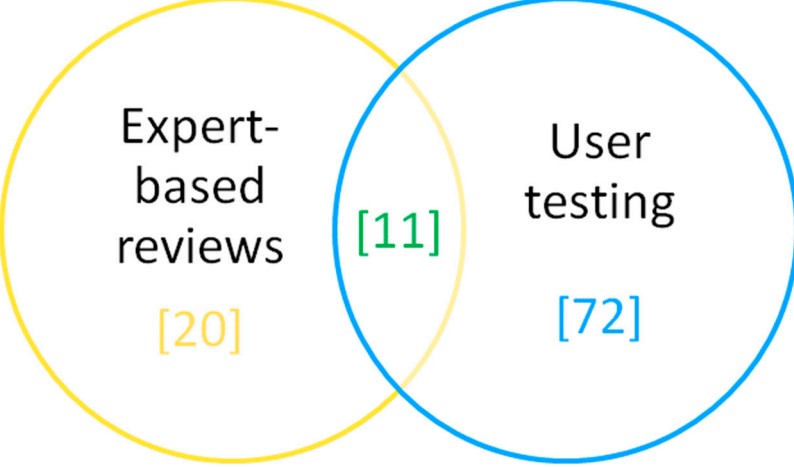

**Figure 4.** Distribution of metrics to specific methods.

**Table 1.** The UXIE framework: concepts, attributes, metrics and methods. Asterisk (*) denotes metrics that are novel in the UXIE framework.

| **Intuitiveness** | | |
|---|---|---|
| Awareness of application capabilities | Functionalities that have been used for each system * | User testing [⏱ D]: Automated measurement |
| | Undiscovered functionalities of each system * | User testing [⏱ D]: Automated measurement |
| Awareness of the interaction vocabulary | Percentage of input modalities used * | User testing [⏱ D]: Automated measurement |
| | Erroneous user inputs (inputs that have not been recognized by the system) for each supported input modality * | User testing [⏱ D]: Automated measurement |
| | Percentage of erroneous user inputs per input modality * | User testing [⏱ D]: Automated measurement |
| **Unobtrusiveness** | | |
| Distraction | Number of times that the user has deviated from the primary task * | User testing [⏱ D] [Task-based, or Think Aloud]: Automation support |
| | Time elapsed from a task deviation until the user returns to the primary task | User testing [⏱ D] [Task-based, or Think Aloud]: Automation support |
| Embedment | The system and its components are appropriately embedded in the surrounding architecture | Expert-based review; User testing [⏱ sA]: Questionnaire, Interview |
| **Adaptability and Adaptivity** | | |
| Input (sensor) data | Accuracy of input (sensor) data perceived by the system | User testing [⏱ D]: Automation support |
| Interpretations | Validity of system interpretations | Expert-based review |
| Appropriateness of adaptation | Interaction modalities are appropriately adapted according to the user profile and context of use * | Expert-based review; User testing [⏱ D]: Automation support |
| | System output is appropriately adapted according to the user profile and context of use * | Expert-based review; User testing [⏱ D]: Automation support |
| | Content is appropriately adapted according to the user profile and context of use * | Expert-based review; User testing [⏱ D]: Automation support |
| | Adaptations that have been manually overridden by the user * | User testing [⏱ D]: Automation support |
| Adaptation impact | Number of erroneous user inputs (i.e., incorrect use of input commands) once an adaptation has been applied * | User testing [⏱ D]: Automated measurement |
| | Number of erroneous user interactions once an adaptation has been applied * | User testing [⏱ D]: Automation support |
| | Percentage of adaptations that have been manually overridden by the user * | User testing [⏱ D]: Automation support |
| Appropriateness of recommendations | The system adequately explains any recommendations | Expert-based review; User testing [⏱ D]: Automation support |
| | The system provides an adequate way for users to express and revise their preferences | Expert-based review |
| | Recommendations are appropriate for the specific user and context of use * | Expert-based review; User testing [⏱ D]: Automation support |
| | Recommendations that have not been accepted by the user * | User testing [⏱ D]: Automation support |
| | Percentage of accepted system recommendations * | User testing [⏱ D]: Automated measurement |
| | User satisfaction by system recommendations (appropriateness, helpfulness/accuracy) | User testing [⏱ sA]: Questionnaire, Interview |

**Table 1.** *Cont.*

| Usability | | |
|---|---|---|
| Conformance with guidelines | The user interfaces of the systems comprising the IE conform to relevant guidelines | Expert-based review |
| Effectiveness | Task success | User testing [🕐 D] (Task-based): Automation support |
| | Number of interaction errors * | User testing [🕐 D]: Automation support |
| | Number of input errors * | User testing [🕐 D]: Automated measurement |
| | Number of system failures | User testing [🕐 D]: Automated measurement |
| Efficiency | Task time | User testing [🕐 D] (Task-based): Automated measurement |
| | Number of help requests | User testing [🕐 D]: Automation support |
| | Time spent on errors | User testing [🕐 D]: Automation support |
| Learnability | Users can easily understand and use the system | Expert-based review (cognitive walkthrough) |
| | Number of interaction errors over time * | User testing [🕐 D]: Automated measurement |
| | Number of input errors over time * | User testing [🕐 D]: Automated measurement |
| | Number of help requests over time * | User testing [🕐 D]: Automated measurement |
| Accessibility | The system conforms to accessibility guidelines | Expert-based review Semi-automated accessibility evaluation tools |
| | The systems of the IE are electronically accessible | Expert review User testing [🕐 D] |
| | The IE is physically accessible | Expert review User testing [🕐 D] User testing [🕐 sA]: Interview |
| Physical UI | The system does not violate any ergonomic guidelines | Expert-based review |
| | The size and position of the system is appropriate for its manipulation by the target user groups | Expert-based review User testing [🕐 D] |
| User satisfaction | Users believe that the system is pleasant to use | User testing [🕐 sA] [🕐 lA]: Questionnaire |
| | Percent of favorable user comments/unfavorable user comments | User testing [🕐 D]: Automation support |
| | Number of times that users express frustration | User testing [🕐 D]: Automation support |
| | Number of times that users express clear joy | User testing [🕐 D]: Automation support |
| Cross-platform usability | After switching device: time spent to continue the task from where it was left * | User testing [🕐 D]: Automation support |
| | After switching device: number of interaction errors until task completion * | User testing [🕐 D]: Automated measurement |
| | Consistency among the user interfaces of the individual systems | Expert-based review |
| | Content is appropriately synchronized for cross-platform tasks | Expert-based review |
| | Available actions are appropriately synchronized for cross-platform tasks | Expert-based review |
| | Help requests after switching devices * | User testing [🕐 D]: Automated measurement |
| | Cross-platform task success compared to the task success when the task is carried out in a single device (per device) * | User testing [🕐 D] (Task-based): Automation support |
| | Cross-platform task time compared to the task time when the task is carried out in a single device (per device) * | User testing [🕐 D] (Task-based): Automated measurement |

**Table 1.** *Cont.*

| | | |
|---|---|---|
| Multiuser usability | Number of collisions with activities of others | User testing [🕐 D]: Automated measurement |
| | Correctness of system's conflict resolution * | User testing [🕐 D]: Automation support |
| | Percentage of conflicts resolved by the system | User testing [🕐 D]: Automated measurement |
| | Percentage of conflicts resolved by the user(s) * | User testing [🕐 D]: Automation support |
| | Social etiquette is followed by the system | Expert-based review |
| Implicit interactions | Implicit interactions carried out by the user * | User testing [🕐 D]: Automated measurement |
| | Number of implicit interactions carried out by the user * | User testing [🕐 D]: Automated measurement |
| | Percentages of implicit interactions per implicit interaction type * | User testing [🕐 D]: Automated measurement |
| | Appropriateness of system responses to implicit interactions * | Expert-based review User testing [🕐 D]: Automation support |
| Usage | Global interaction heat map: number of usages per hour on a daily, weekly and monthly basis for the entire IE * | User testing [🕐 D]: Automated measurement |
| | Systems' interaction heat map: number of usages for IE each system per hour on a daily, weekly and monthly basis * | User testing [🕐 D]: Automated measurement |
| | Applications' interaction heat map: number of usages for each IE application per hour on a daily, weekly and monthly basis * | User testing [🕐 D]: Automated measurement |
| | Time duration of users' interaction with the entire IE * | User testing [🕐 D]: Automated measurement |
| | Time duration of users' interaction with each system of the IE * | User testing [🕐 D]: Automated measurement |
| | Time duration of users' interaction with each application of the IE * | User testing [🕐 D]: Automated measurement |
| | Analysis (percentage) of applications used per system (for systems with more than one application) * | User testing [🕐 D]: Automated measurement |
| | Percentage of systems to which a pervasive application has been deployed, per application * | User testing [🕐 D]: Automated measurement |

**Appeal and Emotions**

| | | |
|---|---|---|
| Aesthetics | The systems follow principles of aesthetic design | Expert-based review |
| | The IE and its systems are aesthetically pleasing for the user | User testing [🕐 sA] [🕐 lA]: Questionnaire |
| Fun | Interacting with the IE is fun | User testing [🕐 sA] [🕐 lA]: Questionnaire |
| Actionable emotions | Detection of users' emotional strain through physiological measures, such as heart rate, skin resistance, blood volume pressure, gradient of the skin resistance and speed of the aggregated changes in the all variables' incoming data | User testing [🕐 D]: Automated measurement |
| | Users' affective reaction to the system | User testing [🕐 sA] [🕐 lA]: Questionnaire |

**Safety and Privacy**

| | | |
|---|---|---|
| User control | User has control over the data collected | Expert-based review |
| | User has control over the dissemination of information | Expert-based review |
| | The user can customize the level of control that the IE has: high (acts on behalf of the person), medium (gives advice), low (executes a person's commands) * | Expert-based review |

**Table 1.** *Cont.*

| | | |
|---|---|---|
| Privacy | Availability of the user's information to other users of the system or third parties | Expert-based review |
| | Availability of explanations to a user about the potential use of recorded data | Expert-based review |
| | Comprehensibility of the security (privacy) policy | Expert-based review |
| Safety | The IE is safe for its operators | Expert-based review |
| | The IE is safe in terms of public health | Expert-based review |
| | The IE does not cause environmental harm | Expert-based review |
| | The IE will not cause harm to commercial property, operations or reputation in the intended contexts of use | Expert-based review |
| **Technology Acceptance and Adoption** | | |
| System attributes | Perceived usefulness | User testing [⏱ sA] [⏱ lA]: Questionnaire |
| | Perceived ease of use | User testing [⏱ sA] [⏱ lA]: Questionnaire |
| | Trialability | Field study/In situ evaluation [⏱ sA]: Questionnaire |
| | Relative advantage | User testing [⏱ sA] [⏱ lA]: Questionnaire |
| | Cost (installation, maintenance) | Field study/In situ evaluation [⏱ sA]: Questionnaire |
| User attributes | Self-efficacy | User testing [⏱ B]: Questionnaire |
| | Computer attitude | User testing [⏱ B]: Questionnaire |
| | Age | User testing [⏱ B]: Questionnaire |
| | Gender | User testing [⏱ B]: Questionnaire |
| | Personal innovativeness | User testing [⏱ B]: Questionnaire |
| Social influences | Subjective norm | User testing [⏱ B]: Questionnaire |
| | Voluntariness | User testing [⏱ B]: Questionnaire |
| Facilitating conditions | End-user support | Field study/In situ evaluation [⏱ sA] [⏱ lA]: Questionnaire |
| | Visibility | Field study/In situ evaluation [⏱ B]: Questionnaire |
| Expected outcomes | Perceived benefit | User testing [⏱ B] [⏱ sA] [⏱ lA]: Questionnaire |
| | Long-term consequences of use | User testing [⏱ B] [⏱ sA] [⏱ lA]: Questionnaire |
| | Observability | User testing [⏱ sA] [⏱ lA]: Questionnaire |
| | Image | User testing [⏱ sA] [⏱ lA]: Questionnaire |
| Trust | User trust towards the system | User testing [⏱ B] [⏱ sA] [⏱ lA]: Questionnaire |

Although the number of metrics to be studied through user testing is large, evaluators will not be required to observe and collect data for all the 83 metrics. In particular, as shown in Figure 5, 30 (36.14%) of these metrics are automatically calculated by the IE, 25 (30.12%) feature automation support, 2 (2.40%) need to be observed manually, 25 (30.12%) will be obtained through subjective methods, and 1 (1.20%) should be acquired through interviews and manual observations. The 26 subjective metrics are proposed to be retrieved by means of interview (1), questionnaires (23), or both questionnaires and interviews (2),

when additional clarifications will be useful towards identifying potential UX problems or specific user attitudes.

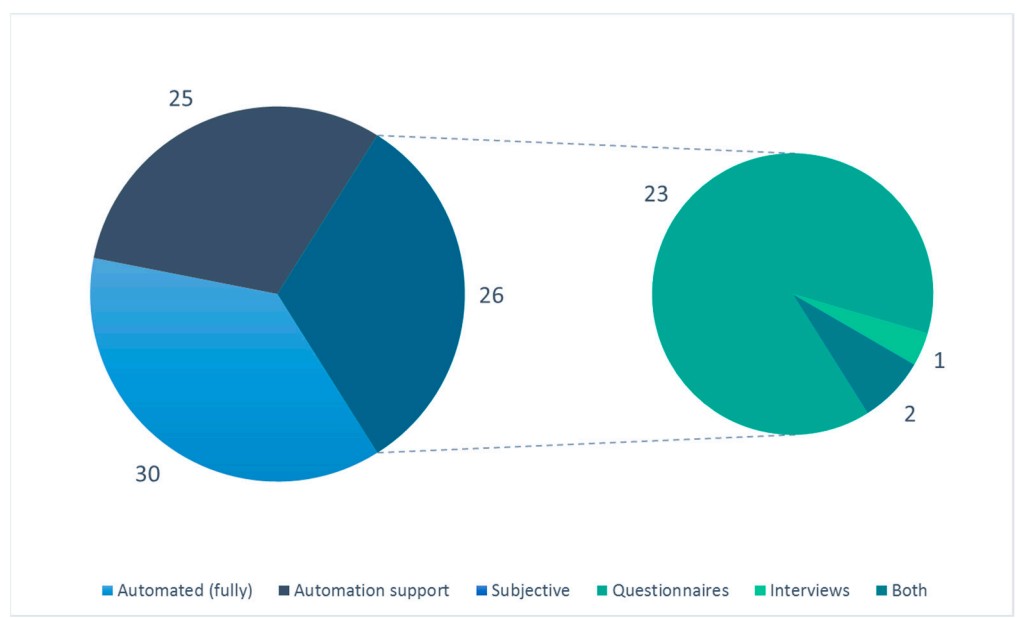

**Figure 5.** Analysis of metrics explored through user testing.

In summary, the UXIE framework proposes that UX evaluation of an intelligent application, system or entire environment should be carried out following a combination of methods and aims at minimizing the number of metrics that should be observed by the evaluator during an evaluation experiment with users. However, the role of experts and evaluators in the process is very significant. It is important to note that human expertise cannot be substituted by any automated evaluation or simulation tool. Instead, these tools aim to provide aggregated metrics, and present them in an appropriate manner in order to facilitate human evaluators in understanding the results and combine them with their own findings and data collected from questionnaires, interviews, or any other usability and UX evaluation methods, so as to effectively comprehend and analyze user experience in an intelligent environment.

## 5. Evaluation of the UXIE Framework

### 5.1. Method and Participants

The proposed framework was evaluated with the participation of six UX practitioners, three of whom were experts in the field, and three knowledgeable. All participants were familiar with the concept of IEs, while three of them had actually carried out a few evaluations of systems operating in IEs in the past. In particular, three of the participants were experts in intelligent systems, having designed and developed systems for more than six years, two were knowledgeable, having less experience as designers of such systems, while one was familiar with such systems, however without any expertise in their design or development. In terms of evaluation of intelligent systems, one participant was expert, having planned and carried out evaluations of such systems for more than four years, two were knowledgeable with two years of active participation in such evaluations, while three were familiar with such evaluations, having participated as observers in a small number (less than five) of such evaluations. Table 2 summarizes the aforementioned data regarding the evaluation participants.

The goal of the evaluation was twofold: (i) assessing if evaluators would plan and carry out a more detailed and inclusive evaluation with the UXIE framework with respect to other methods, and (ii) evaluating the comprehensibility and usability of the framework and retrieving feedback from the evaluators. To this end, the following hypotheses were tested:

**Hypothesis 1 (H1).** *Evaluators will plan a multimethod evaluation with the UXIE framework.*

**Hypothesis 2 (H2).** *The number of metrics that evaluators will examine with the UXIE framework will be larger (compared to the number of metrics that evaluators would plan to measure without the framework).*

**Hypothesis 3 (H3).** *The UXIE framework is usable for evaluators.*

Involving participants who are simply familiar with the concepts of usability and UX, having no practice in actually planning and running evaluations, was considered inappropriate for the context of the current evaluation. Participants should be at least knowledgeable in the field in order to be able to criticize and provide feedback on the framework constructs. Nevertheless, beginner UX practitioners can be involved in future evaluations, where they will be able to use UXIE framework along with tools providing automation support, such as the one reported in [39].

**Table 2.** Evaluation of participants' data.

| | Age | | Usability/UX Expertise | | Evaluation in IEs | |
| --- | --- | --- | --- | --- | --- | --- |
| 20–30 | 2 | Expert | 3 | Expert | 1 |
| 30–40 | 2 | Knowledgeable | 3 | Knowledgeable | 2 |
| 40–50 | 2 | | | Familiar | 3 |

*5.2. Procedure*

A major goal of the evaluation of frameworks is to assess how usable they are for the intended target audience [42], and retrieve qualitative feedback regarding their readability, understandability, learnability, applicability, and usefulness [43]. The evaluation of the proposed framework mainly targeted at retrieving qualitative feedback from evaluators regarding its usability, however, a cognitive exercise was also included in order to retrieve some quantitative metrics as well. More specifically, considering a given scenario the evaluation involved two phases: (a) planning the evaluation without the UXIE framework and (b) planning the same evaluation with the framework. In order to place them in context, an introduction phase preceded, where participants were introduced to their role, being the lead UX expert in the design team of a smart home, whose task is to plan, organize, and carry out evaluations of the systems being developed. In addition, participants were given a specific evaluation target, namely the TV system located in the living room of the smart home, and three short scenarios exemplifying its usage by the home residents. The scenario (given to participants as follows in Table 3) exemplified not only the possible interaction and functionality of the television, but also addressed the topics of implicit interactions, system adaptation, multiuser usage, and system recommendations.

Having been provided with the scenarios, phase A of the evaluation was initiated, during which participants were asked to think and organize the evaluation of the television, noting which methods they would use and what they would measure. They were given two days to think and plan their evaluation. After that, the evaluation method and metrics proposed by each participant were recorded. Following, the UXIE framework was introduced by describing its main purpose, the multimethod approach advocated, the main IE attributes assessed, along with the full or partial automation support proposed in the context of user testing. Moving to phase B of the evaluation, participants were given printouts of the UXIE framework and were asked to read it carefully and think again how they would plan this time the evaluation and also comment on metrics that were not understandable. They were given three days to prepare and plan their evaluation, taking into account that they had to read all the metrics and have the chance to comprehend how the framework works. It should be noted that they were not given a description of what each parameter means, or how important it is in the context of an IE, and why it had been included in the framework. After completing phase B, evaluators' preferred metrics

and comments were recorded. Finally, they were interviewed following a semistructure interview approach featuring the following questions:

1.  What is your overall impression of the UXIE framework?
2.  Would you consider using it? Why?
3.  Was the language clear and understandable?
4.  What was omitted that should have been included?
5.  What could be improved?
6.  Would it be helpful in the context of carrying out evaluations in intelligent environments in comparison to existing approaches you are aware of?

**Table 3.** Evaluation scenario.

| Living Room TV (Interaction: Gestures, Speech, and Remote Control) | |
|---|---|
| Scenario 1 | Jenny enters home after a long day at work. On her way home, she heard on the radio about an earthquake in her home island. Worried, she turns on the TV through the remote control. She switches to her favorite news channel through the remote control and turns up the volume by carrying out a gesture, raising up her palm that faces the ceiling. The news channel is currently showing statements of the Prime Minister for a hot political topic. While listening to the news, she does some home chores and prepares dinner. She is cooking, when she listens that a report about the earthquake is presented and returns to the TV area. It turns out that the earthquake was small after all and no damages have been reported. |
| Scenario 3 | Peter has returned home from work and is currently reading the news through the living room TV. While reading, he receives a message from Jenny that she is on her way home and that he should start the dishwasher. Peter heads towards the kitchen (lights are turned on), selects a dishwasher program to start and returns to the living room (while kitchen lights are automatically turned off). After some time, Jenny arrives at home and unlocks the front door. As Jenny's preferred lighting mode is full bright, while Peter has dimmed the lights, a message is displayed on the active home display, the living room TV, asking whether light status should change to full bright. Peter authorizes the environment to change the lighting mode, welcomes Jenny and they both sit on the couch to read the news. Peter tells Jenny about an interesting article regarding an automobile company and the recent emissions scandal and opens the article for her to read. Having read the article, Jenny recalls something interesting that she read at work about a new car model of the specific company and how it uses IT to detect drivers' fatigue. She returns to the news categories, selects the IT news category and they both look for the specific article. Peter reads it and they continue selecting collaboratively interesting news articles. After some time, and since they have to wait for Arthur—their 15 year old son—to come back from the cinema, they decide to watch a movie. The system recommends movies based on their common interests and preferences. Peter selects the movie, Jenny raises the volume, while the environment dims the lights to the preset mode for watching TV. Quite some time later, and while the movie is close to ending, Arthur comes home. As soon as he unlocks the door and enters, the lights are turned to full bright and the movie stops, since the movie is rated as inappropriate for persons younger than 16 years old. Jenny and Peter welcome their son, and then resume the movie, as they think that it is not inappropriate for Arthur anyway, plus it is about to end. The movie ends and Jenny heads to the kitchen to serve dinner. Arthur and Peter browse through their favorite radio stations and select one to listen to. The dinner is served, the family is gathered in the kitchen, and the music follows along, as it is automatically transferred to the kitchen speaker. |

*5.3. Results*

Analysis of the results of the evaluation revolves around the three hypotheses and explores if and how they are supported.

**Hypothesis 1 (H1).** *Evaluators will plan a multimethod evaluation with the UXIE framework.*

In phase A (prior to using the framework), the following methods were employed for the evaluations that were planned:

- User testing: suggested by all six participants (100%).
- Expert-based reviews: suggested by one participant only (16.66%).

Specifically, in terms of user testing, the following methods were suggested (Figure 6a): observation (6 participants: 100%), questionnaires (5 participants: 83.33%), interview (3 participants: 50%), Experience Recollection Method [44] (1 participant: 16.66%), and UX Graph [44] (1 participant: 16.66%).

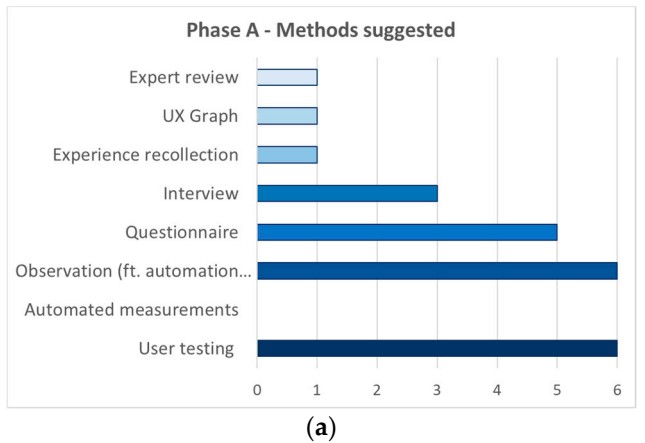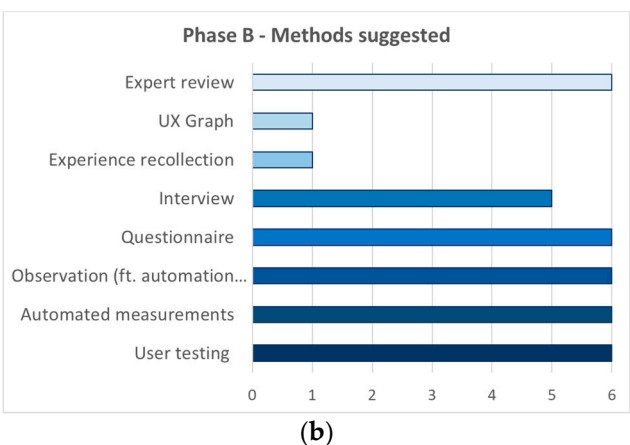

(**a**)           (**b**)

**Figure 6.** Methods suggested for the evaluation of UX in intelligent environments (**a**) without UXIE; (**b**) using UXIE.

In addition, two participants suggested that logs could be used, without however being able to explain how to use them or associate any specific metrics with this method.

In phase B (Figure 6b), all the evaluators selected the expert-based review and the user testing method employing automated measurements, observation through automation support, as well as questionnaires. Interview was selected by five participants, while the methods of Experience Recollection and UX Graph were suggested to be used by the same participant who also employed them in phase A.

By comparing the results acquired in the two phases regarding the methodologies used, the following conclusions hold:

- Although in phase A only one participant selected expert-based reviews as a method to be employed, in phase B six participants selected it, embracing the multimethod approach advocated by the framework.
- Interviews were selected by two more participants in phase B.
- Automated measurements were selected by all the participants in phase B.
- Observations through automation support were selected by all the participants in phase B.

Based on the above, it is evident that hypothesis H1 is supported, as a multimethod approach was selected by all the participants who used the UXIE framework, although without it they had not catered for such a perspective and in their majority had focused on user testing only. Further looking at the metrics selected for each approach, it holds that in phase A only one participant employed expert-based review for a single metric. In phase B however, not only the number of participants suggesting expert-based reviews increased, but also the number of metrics that would be assessed with the use of experts was much higher, leading thus to a more well-balanced iterative approach.

Table 4 provides the number of expert-based review metrics employed by each participant and in average in phases A and B, as well as the percentage of adoption in phase B of the UXIE proposed expert-based review metrics, calculated as per Equation (1), where

$p$ is the number of metrics proposed by the participant and 30 is the total number of expert-based metrics proposed by the framework.

$$(p) = \frac{p}{30},$$ (1)

**Table 4.** Number of expert-based review metrics per evaluation phase.

|  | **P1** | **P2** | **P3** | **P4** | **P5** | **P6** | **Avg.** |
|---|---|---|---|---|---|---|---|
| Phase A | 0 | 0 | 0 | 0 | 0 | 1 | 0.16 |
| Phase B | 3 | 22 | 25 | 16 | 30 | 29 | 20.83 |
| UXIE adoption | 10% | 73.33% | 83.33% | 53.33% | 100% | 96.66% | 69.44% |

**Hypothesis 2 (H2).** *The number of metrics that evaluators will examine with the UXIE framework will be larger.*

In phase A, a total of 46 metrics were proposed by the participants towards measuring UX of the envisioned IE, some of which overlapped. The final list of metrics proposed was:

A.  Observation

   1.  Time to complete a task
   2.  Number of times that an interaction modality is used
   3.  Interaction modality changes for a given task
   4.  Number of errors
   5.  Input errors
   6.  Interaction errors
   7.  Time spent recovering from errors
   8.  Number of help requests
   9.  Number of times that the user "undoes" automatic changes
   10.  Interaction modality accuracy
   11.  Interaction modality selected first
   12.  Task success
   13.  Number of tries to achieve a task
   14.  Unexpected actions or movements
   15.  User confidence with interaction modalities

B.  Think aloud user statements

   16.  Input modalities that the user wanted to use but did not remember how to
   17.  Number of times the user expresses frustration
   18.  Number of times the user expresses joy
   19.  If the user understands the changes happening in the environment

C.  Questionnaires

   20.  Age
   21.  Gender
   22.  Computer attitude
   23.  Preferable interaction technique
   24.  User satisfaction (questionnaire)
   25.  How well did the system manage multiple users?
   26.  Correctness of system adaptations
   27.  Level of fatigue
   28.  Users' experience of the intelligence
   29.  How intrusive did they find the environment?
   30.  Effectiveness (questionnaire)
   31.  Efficiency (questionnaire)
   32.  User feelings

33.  Learnability
34.  System innovativeness
35.  System responsiveness
36.  System predictability
37.  Comfortability with gestures
38.  Promptness of system adaptations to user emotions
39.  Comfortability with tracking and monitoring of activities

D.  Interview

40.  User feedback for each modality
41.  Likes
42.  Dislikes
43.  Additional functionality desired
44.  Experience Recollection Method (ERM)
45.  User experience
46.  UX Graph
47.  User satisfaction from the overall user experience
48.  Expert-based review
49.  Functionality provided for setting preferences

Figure 7a illustrates the number of parameters suggested per participant during phase A. The distribution of the proposed metrics per method is illustrated in Figure 7b, whereby it is evident that 22 metrics (36.66%) pertain to observed user behaviors, 40 metrics (63.49%) are user-reported (i.e., derived through statements vocalized in a think-aloud protocol, questionnaires, or interviews), and 1 metric (1.66%) is based on expert-based reviews. The exact number of proposed metrics per method and per participant is provided in Table 5. In general, it was observed that participants suggested metrics that were reasonable and important in the context of IEs (e.g., preferable interaction technique), however, they typically resorted in measurements through users' self-reporting, with the exception of well-established usability metrics that were suggested to be measured, such as task success, time to complete a task, etc.

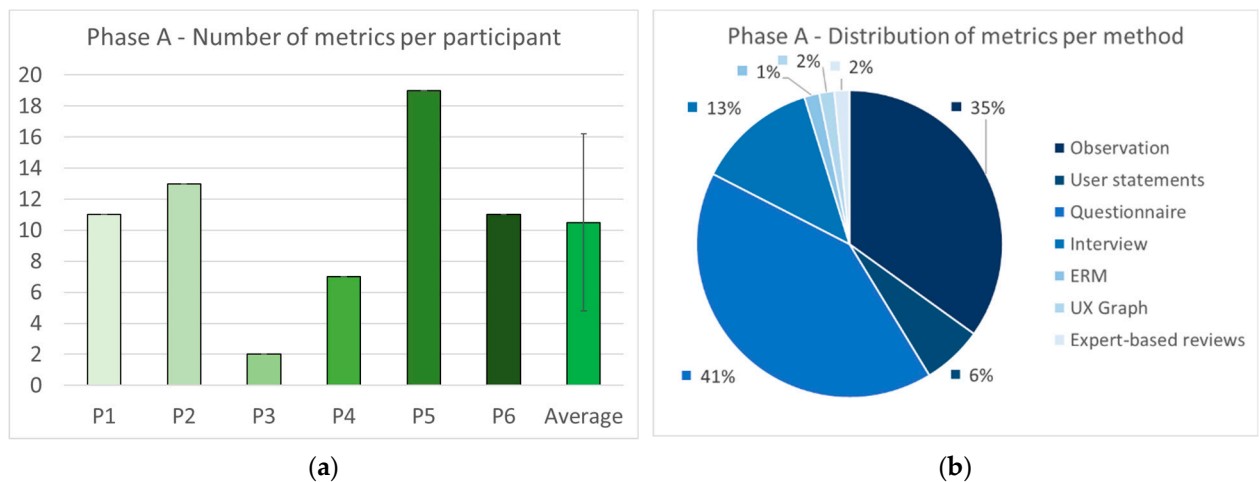

(**a**)                                                           (**b**)

**Figure 7.** Phase A metrics: (**a**) number of metrics per participant (**b**) distribution of metrics per method.

Examining the metrics proposed by evaluators in phase A, from the perspective of the IE attributes and characteristics, it turns out that a small proportion of attributes that should be examined in an IE context was suggested to be included (Figure 8). In particular, the suggested metrics address the issue of User Experience in intelligent environments in a rather low percentage (31.73%). It is noteworthy that certain attributes—although fundamental—are inadequately met, such as privacy and safety (10%), or adaptivity (12.50%) and adoption (26.31%). Moreover, the majority of attributes are only partially

explored, e.g., unobtrusiveness (33.33%), usability (39.13%), as well as appeal and emotions (40%).

**Table 5.** Number of metrics proposed per method in phase A.

| Participant | Observation | User Statements | Questionnaire | Interview | ERM | UX Graph | Expert | Total |
|---|---|---|---|---|---|---|---|---|
| P1 | 7 | 4 | 0 | 0 | 0 | 0 | 0 | 11 |
| P2 | 3 | 0 | 8 | 0 | 1 | 1 | 0 | 13 |
| P3 | 1 | 0 | 0 | 1 | 0 | 0 | 0 | 2 |
| P4 | 4 | 0 | 1 | 2 | 0 | 0 | 0 | 7 |
| P5 | 5 | 0 | 10 | 4 | 0 | 0 | 0 | 19 |
| P6 | 2 | 0 | 7 | 1 | 0 | 0 | 1 | 11 |
| Total | 22 | 4 | 26 | 8 | 1 | 1 | 1 | 63 |

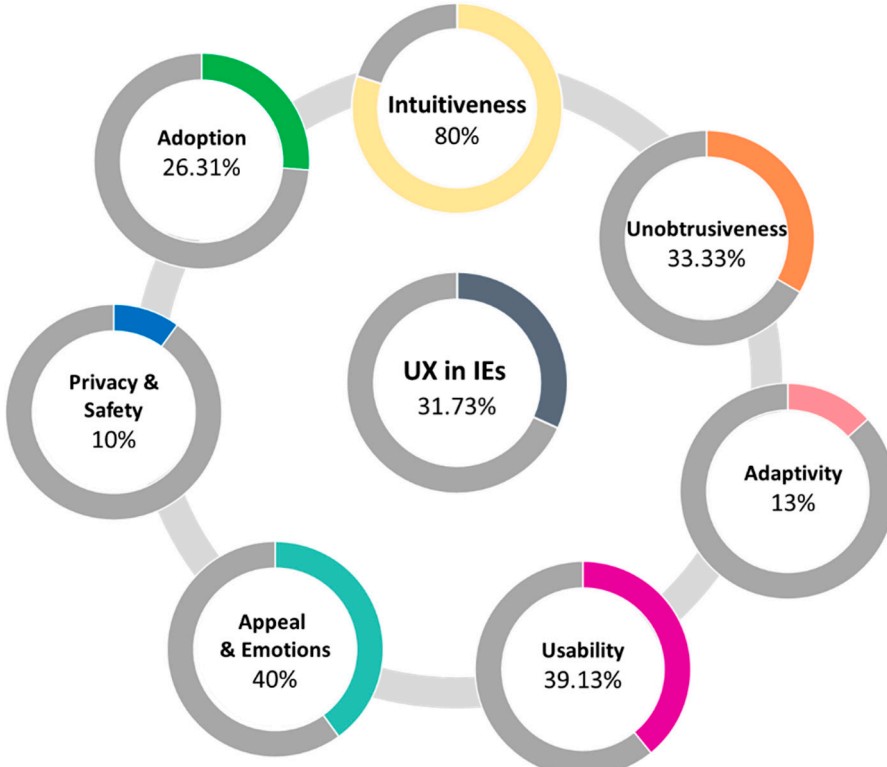

**Figure 8.** Proportion of IE attributes evaluated as suggested by the evaluation participants in phase A.

On the contrary, in phase B, the number of parameters suggested by the participants was considerably larger. Overall, the aggregated number of suggested metrics was 103 (i.e., all the UXIE framework metrics) plus one, namely perceived user experience, as it is quantified through the ERM and UX Graph methods. In phase A, the aggregated number of metrics was 46, it is therefore directly evident that hypothesis H2 is supported, as the number of proposed metrics substantially increased.

Further, besides the aggregated number of metrics, the individual number of metrics per participant also increased considerably, as illustrated in Figure 9. It is notable that the minimum number of metrics was 30 in phase B (P1), whereas the maximum number suggested in phase A was 19 (P5). This increase is clearly demonstrated by the increase in the average number of metrics proposed, which was 10.5 in phase A, against 74 in phase B.

The distribution of metrics to methods has also apparently changed when using the UXIE framework, employing metrics assessed by expert-based reviews, and embracing all the automated metrics. As the entire set of UXIE metrics has been involved in total by all evaluators, the metrics distribution per method is the one advocated by the framework:

31 metrics to be evaluated with expert-based reviews, 30 metrics to be automatically calculated by the IE, 25 to be observed with automation support, 2 to be observed manually, 25 to be obtained through subjective methods, and 1 to be acquired through interviews and manual observations (note that 11 metrics are to be evaluated both by expert-based reviews and user testing methods). Finally, with UXIE all the IE attributes would eventually be assessed in their entirety by the six evaluation participants, in contrast to the extremely partial assessment of phase A.

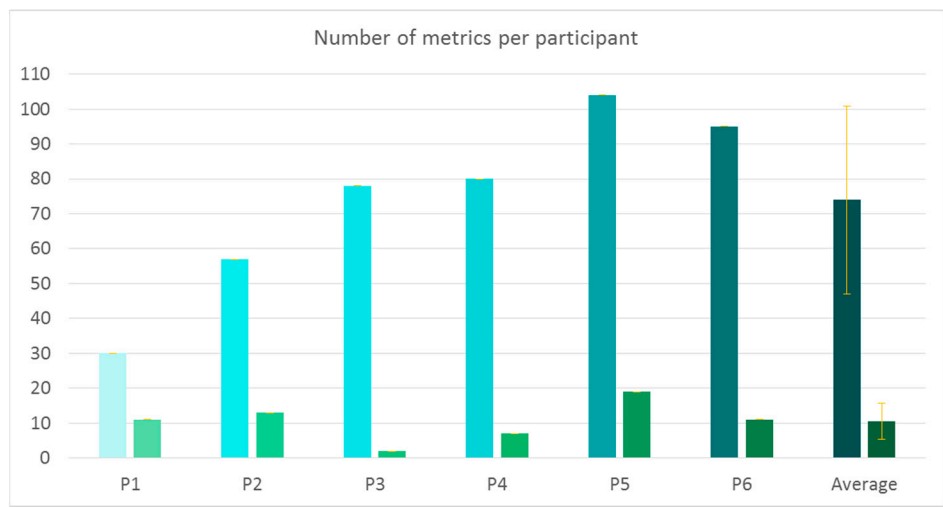

**Figure 9.** Number of metrics suggested per participant in phase B (first column) and phase A (second column).

In conclusion, hypothesis H2 is supported, as the number of metrics that were employed in phase B was greater for each participant individually, in average, and in total.

**Hypothesis 3 (H3).** *The UXIE framework is usable for evaluators.*

To explore this hypothesis the participants' answers provided in the semistructured interview that followed phase B are discussed.

Regarding their overall impression of the framework, participants indicated that it is complete, structured, thorough and in general very good. All participants provided positive answers, an example being the following statement: "A thorough and exhaustive collection of the most important evaluation metrics and heuristics, which is by itself very useful for the evaluator". In terms of understandability, all the participants agreed to the fact that all the metrics were clear and understandable, with the exception of certain specific metrics pertaining to Technology Acceptance. However, as one of them clarified, it only required a few minutes to refresh their memory of what these metrics mean by looking into the related theories. It should be mentioned that with the goal to assess how intuitive the metrics are, evaluators were not given any explanation or accompanying material regarding the metrics. To resolve this issue, a short list with terms and definitions was prepared (Appendix A), which will accompany the UXIE framework.

Evaluators were also asked what was omitted from the framework. Regarding omissions, all the evaluators agreed that they could not find any metrics or aspects of IE environments missing. Some evaluators suggested employing expert-based reviews along with user-based testing for four specific metrics. Their suggestions were adopted and have already been included in the framework. One evaluator highlighted the need for being directed towards which questionnaires to employ, with an emphasis on standardized ones. Although the initial intention of the UXIE framework was to allow evaluators to employ any specific user testing method and protocol, as well as any questionnaires they prefer, this suggestion will be adopted in future versions of the framework, which will include

potentially useful questionnaires that could be used, without, however, being imperative for evaluators to adopt them.

Regarding improvements, the majority of evaluators suggested that the framework could be accompanied by tools to facilitate automated measurements and inspections. Such a tool has already been developed and is reported in [39], while other similar tools are expected to appear in the near future. Furthermore, half of the evaluators suggested that they would have liked to have distinct tables for each method. The approach of one unified table was initially preferred, as on the one hand it provides an overview of all the metrics that fall under a specific attribute of the IE, and on the other hand it makes clear that some metrics can and should be evaluated in a multimethod approach. The suggestion was, however, adopted, and the framework will be accompanied further distinct tables (as provided in Appendix B), providing different classifications of the metrics, and namely metrics that should be assessed through expert-based reviews, questionnaire-based metrics for user-based experiments to be acquired before the experiment, observation metrics that can be automatically acquired with the help of the IE during the experiment, observation metrics regarding the experiment that need to be marked by the evaluator and receive automation support for calculations through tools, metrics that should be pursued through questionnaires or interviews shortly after the system usage in a user-based experiment, as well as metrics that should be acquired a long time after the system usage through questionnaires. Urged by the same need of easily retrieving metrics per method, two evaluators suggested that an electronic version of the framework, offering filters and step-by-step guides would also be useful, an observation that will be certainly followed up in future work.

Finally, evaluators were asked if they would consider actually using the framework and how helpful they think it would be in the context of evaluations in IEs. All responses were unanimous, highlighting that they would definitely use the framework in any evaluation (not only IE oriented), as it is thorough, systematic, well-structured, "a real problem solver". In addition, it was stressed that using the framework will reduce the time required for preparing and running an evaluation, and that one of its major benefits is that it minimizes the need for long questionnaires and lengthy interviews and substitutes them with actually measurable behaviors. Especially with regard to IEs, evaluators pointed out that it is the first framework that they know of specifically targeted to IEs, therefore it outweighs existing approaches. Further, the automated measurements it suggests are highly valuable and make it possible to collect data otherwise impossible to retrieve.

Based on the above analysis of evaluators' responses in the interview, it can be concluded that H3 is supported and that the UXIE framework is not only usable, but actually useful and valuable for evaluators.

## 6. Discussion

According to the analysis of hypotheses H1 and H2, it turns out that evaluators employed a more balanced approach in phase B, where by using the UXIE framework they were able to avoid estimations based entirely on user-reported perceptions and moved towards metrics objectively assessed through the environment itself, or through observed behaviors analyzed systematically with the potential assistance of tools (automation support). It is also evident that they all realized the importance of expert-based reviews and decided to adopt an iterative evaluation approach, gaining all the benefits it promises. Moreover, with the help of the framework, evaluators were able to plan a more thorough evaluation of user experience, based on metrics beyond the typical ones employed in usability evaluations (e.g., errors or task success), and to incorporate attributes of IEs that would have been otherwise neglected. Moreover, using the UXIE framework, the evaluation catered for all the temporal facets of UX, namely before, during, shortly after, and long after using a product.

Analysis of participants' responses also led to identifying additional UX parameters that should be employed in future versions of the UXIE framework. These parameters are:

- Level of fatigue, which is an important consideration for the evaluation of systems supporting gestures. Although context-specific, as gestures are expected to be a fundamental interaction modality in IEs, this metric will be included in future versions of the UXIE framework, along with other metrics examining the most fundamental interaction modalities. In addition, such specific concerns are expected to be studied by expert evaluators as well.
- System responsiveness: a system characteristic which obviously impacts the overall user experience that should always be examined during software testing. Future versions of the framework will consider adding this variable to the expert-based measurements and simulations, however, not as a user reported metric.
- Comfortability with tracking and monitoring of activities: a fundamental concern in IEs is whether users accept the fact that the environment collects information based on their activities. UXIE has included attributes regarding safety, privacy, and user's control over the behavior of the IE. In addition, the trust metric in the acceptance and adoption category aims to retrieve users' attitudes on how much they trust the IE. Future versions of the framework will explore if a specific question for activity monitoring should be included as well.
- Perceived overall user experience: this metric could be included as an indication that additional methods estimating user experience as perceived by the users can be employed (e.g., how satisfied they are from the system during the various phases of using it).

Overall, through the analysis of the evaluators' interview responses, it can be affirmed that the UXIE framework is understandable, detailed, complete, and well-structured. All the evaluators acknowledged its usefulness towards any evaluation and highlighted its innovativeness in terms of evaluation in intelligent environments. The usage of automated measurements was emphatically appraised, along with other benefits of the framework, such as that it provides a complete guide, facilitating evaluators in planning and carrying out thorough evaluations in a more "standardized" manner with minimum time required for preparation.

## 7. Conclusions

This paper presented UXIE, a framework for evaluating User Experience in Intelligent Environments, which aims to become a useful tool for UX evaluation experts towards designing and evaluating intelligent systems, applications and entire environments. Taking into account best practices in the literature, and more specifically approaches for the evaluation of adaptive systems, UbiComp systems, as well as for UX and usability evaluation and technology acceptance, the proposed framework introduces a holistic approach that can be applied in any context of use. Given the complexity of IEs, and the wide range of potential contexts and target users, the framework does not constitute a panacea for any potential system; instead, it is an extensible approach taking into account the various attributes of IEs and parameters of interaction. It aims to provide a solid and clean-cut basis for the UX evaluation in any IE, which can be further augmented with context-specific metrics if needed (e.g., metrics related to enhanced visitor flow in an intelligent museum, support of medical practices in an intelligent hospital, etc.).

The UXIE framework constitutes both a conceptual and a methodological tool, describing not only attributes that should be measured, but concrete metrics as well, along with suggestions on the methods to be used towards acquiring the specified metrics. A challenge towards the development of the framework was the immense number of parameters that should be studied, given the complexity and multidimensionality of IEs, as well as the different temporal dimensions of UX and its multiple facets. As a result, an important concern that has guided the development of the framework was the trade-off between a huge list of metrics that would probably cover every possible aspect of an intelligent system and the practical applicability of the framework in real contexts. To this end, the framework foresees the evaluation of an intelligent system/environment through different

phases and supports both formative and summative evaluations. The UX practitioner is therefore provided with a consolidated, easy to manage list of metrics for each evaluation approach/phase.

Taking advantage of the infrastructure of IEs [31,45], UXIE identifies a number of metrics and parameters that can be automatically calculated during a user testing session, alleviating the need for observers to explicitly record them. At the same time, this inherent support by the IE provides an alternative to the common practice of asking users about almost everything, ending up with very lengthy questionnaires, requiring much time to answer and administer. Besides facilitating evaluators and users, the approach of automatically calculating metrics constitutes the missing link in mismatches and gaps often noticed in observers' recordings and users' questionnaire responses.

Another important concern for the development of the framework was to encompass best practices for the evaluation of intelligent environments and to support both short-term and long-term evaluations with real users in simulation spaces (Living Labs) and facilitate practitioners in employing the appropriate metrics for each experiment type. For instance, in the case of short-term task-based evaluations, it is straightforward and meaningful to calculate task success, whereas this is almost impossible in situations where users are instructed to use the environment at their own discretion without a specific scenario. Towards this direction, UXIE not only indicates the method to be applied (i.e., user testing), but also specifies the experiment type for which a metric is better suited. By the same token of guiding evaluators to apply the framework, a clear distinction of the attributes that should be measured along the different temporal dimensions of UX (i.e., before use, during use, shortly after use, long-term after use) is made in the case of UX experiments.

In addition, two significant research directions that are recognized and embraced by the UXIE framework are technology acceptance theories and models, as well as the layered evaluation approach. With regard to the first, common practice so far has been to assess every aspect of the user's attitude through questionnaires, in order to calculate and predict the acceptance of a given technology. UXIE provides a new means for substituting user-provided metrics related to one's experience with the system with observed and automatically calculated metrics. At the same time, it includes metrics stemming from users themselves, reflecting thereby their opinions, with a clear indication on when they should be measured according to the temporal UX dimensions. Concerning the layered evaluation approach, UXIE has adopted the suggestion that an appropriate adaptation is a result of correct input data, valid inferences, and suitable instantiation of the adaptation itself, and guides evaluators towards assessing each of the above separately.

Overall, the proposed framework espousing the notion that user experience is unique for each individual and that it is not a matter of simply adhering to specific guidelines, aims at constituting a tool for evaluators and designers to identify potential UX problems and eliminate them, by adopting a multimethod evaluation approach. Evaluation results have indicated that it is indeed a very useful tool for the evaluation of UX, which can empower researchers and practitioners toward well-planned, coherent, and complete studies throughout the design and development lifecycle of an intelligent system, application, or environment.

Reflecting on the limitations of the framework, as these were also revealed through the evaluation, two main concerns can be identified. First, the framework may seem overwhelming, especially for novice evaluators, who may encounter difficulties in selecting the methods that should be followed and the particular metrics to be employed. In this respect, the framework was extended to include core definitions and tables indicating which metrics should be employed per evaluation method. An additional solution to this limitation is an electronic version of the framework itself, assisting evaluators through wizards to select the most appropriate evaluation method and metrics to employ, according to the implementation status of the system, application, or IE they wish to evaluate, as well as the available resources. One additional concern refers to the actual undertaking of the evaluation study and reporting of results. Although a tool for user-based evaluations

has already been developed and reported in literature, additional tools are needed for expert-based reviews, in order to assist evaluators in selecting the appropriate evaluation guidelines that are suitable and should be examined according to the evaluation target. The development of such tools also constitutes a work that will be pursued in the near future.

Besides any frameworks and tools, however, in order to truly advance state-of-the-art in the field of UX evaluation in the newfangled intelligent environments, and the future AI-empowered technological environments that will come, practitioners would benefit from access to an online network of peers, acting as a resource for best practices and knowledge. This can be facilitated through an online community for UX researchers and practitioners, empowered by content and expertise contributed by its members, but also acting as one's personal repository and single point of access to the aforementioned tools, namely electronic version of the framework, as well as tools for planning and carrying out expert-based and user-based evaluations. Such an approach would also ensure that the framework and evaluation tools are continuously updated and improved, in order to serve the needs of the target audience in the best possible way. Future endeavors will target at developing and promoting such an online collaboration environment, aiming to make headway in the field of UX assessment in contemporary and future technological environments.

**Author Contributions:** Conceptualization, S.N. and C.S.; methodology, S.N.; validation, S.N. and M.A.; formal analysis, S.N. and G.M.; investigation, S.N. and G.M.; writing—original draft preparation, S.N and G.M.; writing—review and editing, S.N., M.A. and C.S.; visualization, S.N.; funding acquisition, C.S. All authors have read and agreed to the published version of the manuscript.

**Funding:** This research work was supported by the FORTH-ICS internal RTD Programme 'Ambient Intelligence'.

**Institutional Review Board Statement:** Not applicable.

**Informed Consent Statement:** Not applicable.

**Data Availability Statement:** Data is contained within the article.

**Acknowledgments:** The authors would like to thank all evaluators for their invaluable comments. Thank you also to anonymous reviewers for constructive criticism.

**Conflicts of Interest:** The authors declare no conflict of interest.

## Appendix A

This appendix provides a list of definitions for terms employed by the UXIE framework, in order of appearance in the framework.

**Accuracy of input (sensor) data perceived by the system**: Check how accurate the raw data received by the system are (e.g., data from sensors).

**Validity of system interpretations**: Check how valid the meaning given by the system to the collected raw data is.

**Input error**: an error referring to incorrect usage of input modalities.

**Interaction error**: an error referring to incorrect usage of the user interface (e.g., selecting an inappropriate menu item for the task at hand).

**Cross-platform task**: a task that is carried out in more than one device.

**Correctness of system's conflict resolution**: Assessment of how correct the decision taken by the system to resolve a conflict of interests/demands between two or more users was.

**Social etiquette**: code of behavior that delineates expectations for social behavior according to contemporary conventional norms within a society, social class, or group.

**Implicit interaction**: an action performed by the user that is not primarily aimed to interact with a computerized system but which such a system understands as input.

**Systems to which a pervasive application has been deployed**: pervasiveness refers to the capability of an application to run in multiple systems (e.g., tablet, smartphone, large screen display).

**Trialability**: the degree to which an innovation may be experimented with before adoption.

**Relative advantage**: the degree to which an innovation is perceived as being better than its precursor/competitors.

**Self-efficacy**: an individual's convictions about his or her abilities to mobilize motivation, cognitive resources and courses of action needed to successfully execute a specific task within a given context (e.g., computer self-efficacy is defined as an individual judgement of one's capability to use a computer).

**Personal innovativeness**: the individual's willingness to try out any new technology

**Subjective norm**: a person's perception that most people who are important to them think the person should or should not perform the behavior in question.

**Voluntariness**: the extent to which potential adopters perceive the adoption decision to be non-mandatory.

**End-user support**: specialized instruction, guidance, coaching and consulting.

**Visibility**: the degree to which the innovation is visible in context of use (e.g., the organization).

**Observability**: the degree to which aspects of an innovation may be conveyed to others.

**Image**: the degree to which use of an innovation is perceived to enhance one's status in one's social system.

**Appendix B**

Classification of metrics per method and per UX evaluation phase: metrics that should be assessed through expert-based reviews (Table A1), questionnaire-based metrics for user-based experiments to be acquired before the experiment (Table A2), observation metrics that can be automatically acquired with the help of the IE during the experiment (Table A3), observation metrics regarding the experiment that need to be marked by the evaluator and receive automation support for calculations through tools (Table A4), metrics that should be pursued through questionnaires (Table A5) or interviews (Table A6) shortly after the system usage in a user-based experiment, as well as metrics that should be acquired long time after the system usage through questionnaires (Table A7).

**Table A1.** UXIE metrics to be assessed through expert-based reviews.

| **Unobtrusiveness** | |
| --- | --- |
| Embedment | The system and its components are appropriately embedded in the surrounding architecture |
| **Adaptability and Adaptivity** | |
| Interpretations | Validity of system interpretations |
| Appropriateness of adaptation | Interaction modalities are appropriately adapted according to the user profile and context of use |
| | System output is appropriately adapted according to the user profile and context of use |
| | Content is appropriately adapted according to the user profile and context of use |
| Appropriateness of recommendations | The system adequately explains any recommendations |
| | The system provides an adequate way for users to express and revise their preferences |
| | Recommendations are appropriate for the specific user and context of use |
| **Usability** | |
| Conformance with guidelines | The user interfaces of the systems comprising the IE conform to relevant guidelines |
| Learnability | Users can easily understand and use the system |

**Table A1.** *Cont.*

| | |
|---|---|
| Accessibility | The system conforms to accessibility guidelines |
| | The systems of the IE are electronically accessible |
| | The IE is physically accessible |
| Physical UI | The system does not violate any ergonomic guidelines |
| | The size and position of the system is appropriate for its manipulation by the target user groups |
| Cross-platform usability | Consistency among the user interfaces of the individual systems |
| | Content is appropriately synchronized for cross-platform tasks |
| | Available actions are appropriately synchronized for cross-platform tasks |
| Multiuser usability | Social etiquette is followed by the system |
| Implicit interactions | Appropriateness of system responses to implicit interactions |
| **Appeal and emotions** | |
| Aesthetics | The systems follow principles of aesthetic design |
| **Safety and privacy** | |
| User control | User has control over the data collected |
| | User has control over the dissemination of information |
| | The user can customize the level of control that the IE has: high (acts on behalf of the person), medium (gives advice), low (executes a person's commands) |
| Privacy | Availability of the user's information to other users of the system or third parties |
| | Availability of explanations to a user about the potential use of recorded data |
| | Comprehensibility of the security (privacy) policy |
| Safety | The IE is safe for its operators |
| | The IE is safe in terms of public health |
| | The IE does not cause environmental harm |
| | The IE will not cause harm to commercial property, operations or reputation in the intended contexts of use |

**Table A2.** UXIE metrics to be assessed through questionnaires before a user-based study.

| **Technology Acceptance and Adoption** | |
|---|---|
| User attributes | Self-efficacy |
| | Computer attitude |
| | Age |
| | Gender |
| | Personal innovativeness |
| Social influences | Subjective norm |
| | Voluntariness |
| Facilitating conditions | Visibility |
| Expected outcomes | Perceived benefit |
| | Long-term consequences of use |
| Trust | User trust towards the system |

**Table A3.** UXIE metrics automatically measured during a user-based study.

| **Intuitiveness** | |
| --- | --- |
| Awareness of application capabilities | Functionalities that have been used for each system |
| | Undiscovered functionalities of each system |
| Awareness of the interaction vocabulary | Percentage of input modalities used |
| | Erroneous user inputs (inputs that have not been recognized by the system) for each supported input modality |
| | Percentage of erroneous user inputs per input modality |
| **Adaptability and adaptivity** | |
| Adaptation impact | Number of erroneous user inputs (i.e., incorrect use of input commands) once an adaptation has been applied |
| Appropriateness of recommendations | Percentage of accepted system recommendations |
| **Usability** | |
| Effectiveness | Number of input errors |
| | Number of system failures |
| Efficiency | Task time |
| Learnability | Number of interaction errors over time |
| | Number of input errors over time |
| | Number of help requests over time |
| Cross-platform usability | After switching device: number of interaction errors until task completion |
| | Help requests after switching devices |
| | Cross-platform task time compared to the task time when the task is carried out in a single device (per device) |
| Multiuser usability | Number of collisions with activities of others |
| | Percentage of conflicts resolved by the system |
| Implicit interactions | Implicit interactions carried out by the user |
| | Number of implicit interactions carried out by the user |
| | Percentages of implicit interactions per implicit interaction type |
| Usage | Global interaction heat map: number of usages per hour on a daily, weekly and monthly basis for the entire IE |
| | Systems' interaction heat map: number of usages for each system in the IE per hour on a daily, weekly and monthly basis |
| | Applications' interaction heat map: number of usages for each application in the IE per hour on a daily, weekly and monthly basis |
| | Time duration of users' interaction with the entire IE |
| | Time duration of users' interaction with each system of the IE |
| | Time duration of users' interaction with each application of the IE |
| | Analysis (percentage) of applications used per system (for systems with more than one application) |
| | Percentage of systems to which a pervasive application has been deployed, per application |
| **Appeal and emotions** | |
| Actionable emotions | Detection of users' emotional strain through physiological measures, such as heart rate, skin resistance, blood volume pressure, gradient of the skin resistance and speed of the aggregated changes in the all variables' incoming data |

**Table A4.** UXIE metrics that should be measured during a user-based study and can have automation support by tools.

| **Unobtrusiveness** | |
|---|---|
| Distraction | Number of times that the user has deviated from the primary task |
| | Time elapsed from a task deviation until the user returns to the primary task |
| **Adaptability and Adaptivity** | |
| Input (sensor) data | Accuracy of input (sensor) data perceived by the system |
| Interpretations | Validity of system interpretations |
| Appropriateness of adaptation | Interaction modalities are appropriately adapted according to the user profile and context of use |
| | System output is appropriately adapted according to the user profile and context of use |
| | Content is appropriately adapted according to the user profile and context of use |
| | Adaptations that have been manually overridden by the user |
| Adaptation impact | Number of erroneous user interactions once an adaptation has been applied |
| | Percentage of adaptations that have been manually overridden by the user |
| Appropriateness of recommendations | The system adequately explains any recommendations |
| | Recommendations are appropriate for the specific user and context of use |
| | Recommendations that have not been accepted by the user |
| **Usability** | |
| Effectiveness | Task success |
| | Number of interaction errors |
| Efficiency | Number of help requests |
| | Time spent on errors |
| User satisfaction | Percent of favourable user comments/unfavorable user comments |
| | Number of times that users express frustration |
| | Number of times that users express clear joy |
| Cross-platform usability | After switching device: time spent to continue the task from where it was left |
| | Cross-platform task success compared to the task success when the task is carried out in a single device (per device) |
| Multiuser usability | Correctness of system's conflict resolution |
| | Percentage of conflicts resolved by the user(s) |
| Implicit interactions | Appropriateness of system responses to implicit interactions |

**Table A5.** UXIE metrics to be assessed through questionnaires shortly after a user-based study.

| | |
|---|---|
| **Unobtrusiveness** | |
| Embedment | The system and its components are appropriately embedded in the surrounding architecture |
| **Adaptability and adaptivity** | |
| Appropriateness of recommendations | User satisfaction by system recommendations (appropriateness, helpfulness/accuracy) |
| **Usability** | |
| User satisfaction | Users believe that the system is pleasant to use |
| **Appeal and emotions** | |
| Aesthetics | The IE and its systems are aesthetically pleasing for the user |
| Fun | Interacting with the IE is fun |
| Actionable emotions | Users' affective reaction to the system |
| **Technology acceptance and adoption** | |
| System attributes | Perceived usefulness |
| | Perceived ease of use |
| | Trialability |
| | Relative advantage |
| | Cost (installation, maintenance) |
| Facilitating conditions | End-user support |
| Expected outcomes | Perceived benefit |
| | Long-term consequences of use |
| | Observability |
| | Image |
| Trust | User trust towards the system |

**Table A6.** UXIE metrics to be assessed through interviews shortly after a user-based study.

| | |
|---|---|
| **Unobtrusiveness** | |
| Embedment | The system and its components are appropriately embedded in the surrounding architecture |
| **Adaptability and adaptivity** | |
| Appropriateness of recommendations | User satisfaction by system recommendations (appropriateness, helpfulness/accuracy) |
| **Usability** | |
| Accessibility | The IE is physically accessible |

**Table A7.** UXIE metrics to be assessed through questionnaires long after a user-based study.

| **Usability** | | |
|---|---|---|
| User satisfaction | Users believe that the system is pleasant to use | |
| **Appeal and emotions** | | |
| Aesthetics | The IE and its systems are aesthetically pleasing for the user | |
| Fun | Interacting with the IE is fun | |
| Actionable emotions | Users' affective reactions to the system | |
| **Technology acceptance and adoption** | | |
| System attributes | Perceived usefulness | |
| | Perceived ease of use | |
| | Relative advantage | |
| Facilitating conditions | End-user support | |
| Expected outcomes | Perceived benefit | |
| | Long-term consequences of use | |
| | Observability | |
| | Image | |
| Trust | User trust toward the system | |

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
