# Peer review of "User Experience Evaluation in Intelligent Environments: A Comprehensive Framework"

_technologies, doi:10.3390/technologies9020041_

Round 1

Reviewer 1 Report

no further comments

Author Response

We would like to thank the reviewer for their time and effort in reading our manuscript.

Reviewer 2 Report

This paper presents a methodological and conceptual framework that provides concrete guidance for UX research, design, and evaluation, explaining which UX parameter should be measured, how, and when. The novel comprehensive framework, named UXIE, assesses a wide range of characteristics and qualities of intelligent environments (IEs), taking into account traditional and modern models and evaluation approaches.

UXIE bridges existing frameworks, provides concrete guidance for the evaluation of UX in intelligent environments and offers methodological guidance. Checklists can be used by evaluators to identify which what to measure and when.

The paper is very coherent, original, and rich in data presentation from a scientific and technical perspective, however correctly analyzed. This work contributes to the advancing state of the art and is then of high interest. It is worth being published.

Author Response

We would like to thank the reviewer for their time and effort in reading our manuscript, and express our gratitude for the positive and encouraging review remarks.

Reviewer 3 Report

This technical report for questionaire design for User Experience Evaluation in Intelligent Environments is not suitable for publication as a technical paper. It is long and verbose, and there is also no result of implementation. 

Author Response

We would like to thank the reviewer for their time and effort in reading our manuscript.

Kindly note that the work described does not propose a questionnaire or a technical tool. It is a research article proposing a conceptual and methodological framework for User Experience evaluation in Intelligent Environments. Its main goal is to guide UX researchers in deciding what to evaluate and which method to follow. The revised manuscript has been updated to clearly indicate this and communicate more clearly the purpose and the innovative aspects of this work to readers.

Reviewer 4 Report

Paper deals with important task. Authors have proposed a new UXIE framework, for the evaluation of User Experience in intelligent environments 

Paper has great practical value.

It has a logical structure, all necessary sections. 

The proposed approach are logical, results are clear.

Suggestions:

  1. Introduction section should be extended using more clearly motivation of this paper.
  2. It would be good to add clear, point-by-point the main contributions in the end of the Introduction section
  3. It would be good to add the remainder of this paper
  4. Related works section should be extended using other works, for example  10.1016/j.cirpj.2021.01.001
  5. Fig 6 and 7 are very small. Please fix it.
  6. Conclusion section should be extended using: limitations of the proposed framework and prospects for the future research.
  7. Some of references are outdated. Please fix it using 3-5 years old papers in high-impact journals.

Author Response

We thank the reviewer for their time and effort in reading our manuscript and for providing constructive review comments.

Please find below our point-by-point responses to the concerns raised:

Introduction section should be extended using more clearly motivation of this paper.

We have extended the introduction section to elaborate on the motivations of this paper, highlighting the challenges imposed to UX evaluation by intelligent environments, and the lack of systematic approaches to address them.

It would be good to add clear, point-by-point the main contributions in the end of the Introduction section

We have added point-by-point the main contributions of this work in the end of the introduction section, right above the  paragraph introducing the structure of the paper. Furthermore, in the framework itself, all the novel metrics have now been clearly indicated.

It would be good to add the remainder of this paper

Thank you! It has been added.

Related works section should be extended using other works, for example  10.1016/j.cirpj.2021.01.001

Related section discusses usability and user experience evaluation frameworks, elaborating also on frameworks specially focused on (ambient) intelligent and smart environments, as well as pervasive and ubiquitous environments, as the ones more relevant to the proposed framework. Additional references have been added to address this comment, as well as comment 7.

Fig 6 and 7 are very small. Please fix it.

They have been updated, using larger font size so that they are easier to read.

Conclusion section should be extended using: limitations of the proposed framework and prospects for the future research.

Thank you for highlighting this important omission. We have extended the conclusion section accordingly to discuss limitations and prospects for future research.

Some of references are outdated. Please fix it using 3-5 years old papers in high-impact journals.

We have added seven more recent papers in the references section, dated after 2015 and extended accordingly the related work discussion, addressing review comment 2.

Again, we would like to thank the reviewer for their valuable insights and suggestions.